# Physics-Informed Neural Operator for Learning Partial Differential Equations

## Abstract

Machine learning methods have recently shown promise in solving partial differential equations (PDEs). They can be classified into two broad categories: solution function approximation and operator learning. The Physics-Informed Neural Network (PINN) is an example of the former while the Fourier neural operator (FNO) is an example of the latter. Both these approaches have shortcomings. The optimization in PINN is challenging and prone to failure, especially on multi-scale dynamic systems. FNO does not suffer from this optimization issue since it carries out supervised learning on a given dataset, but obtaining such data may be too expensive or infeasible. In this work, we propose the physics-informed neural operator (PINO), where we combine the operating-learning and function-optimization frameworks, and this improves convergence rates and accuracy over both PINN and FNO models. In the operator-learning phase, PINO learns the solution operator over multiple instances of the parametric PDE family. In the test-time optimization phase, PINO optimizes the pre-trained operator ansatz for the querying instance of the PDE. Experiments show PINO outperforms previous ML methods on many popular PDE families while retaining the extraordinary speed-up of FNO compared to solvers. In particular, PINO accurately solves long temporal transient flows and Kolmogorov flows, while PINN and other methods fail to converge.

## 1 Introduction

Machine learning-based methods are starting to show promise in scientific computing and especially in solving partial differential equations (PDEs). They have demonstrated advantages in both efficiency and accuracy compared to conventional solvers. They are even able to tackle previously intractable problems such as higher-dimensional, multi-scale, high-contrast, and chaotic PDE systems (Um et al., 2020; Brunton et al., 2020; Fan et al., 2018; Long et al., 2018; Han et al., 2018; Bruno et al., 2021). Broadly, ML-based approaches for PDEs can be divided into two categories: optimizing to solve for a specific solution function of PDE vs. learning the solution operator over a family of PDEs.

**Optimization of solution function and PINN.** Most ML-based methods, as well as the conventional solvers, fall into this category. Conventional solvers such as FDM and FEM usually discretize the domain into a grid and optimize/approximate the solution function on the grid, which imposes a truncation error. The Physics-Informed Neural Network (PINN)-type methods are proposed to overcome the discretization issue (Raissi et al., 2019). They use a neural network as the ansatz of the solution function and take advantage of auto-differentiation to compute the exact, mesh-free derivatives. Recently, researchers have developed numerous variations of PINN with promising results on inverse problems and partially observed tasks (Lu et al., 2021a; Zhu et al., 2019; Smith et al., 2021). However, compared to conventional solvers, PINNs face several optimization issues: (1) the challenging optimization landscape from soft physics or PDE constraints (Wang et al., 2021a), (2) the difficulty to propagate information from the initial or boundary conditions to unseen parts of the interior or to future times (Dwivedi & Srinivasan, 2020), and (3) the sensitivity to hyper-parameters selection (Sun et al., 2020). As a result, PINNs are still unable to compete with conventional solvers in most cases, and they often fail to converge on high-frequency or multi-scale PDEs (Wang et al., 2020b; Fuks & Tchelepi, 2020; Raissi et al., 2020). In this work, we propose to overcome these optimization challenges by integrating operator learning with PINN.

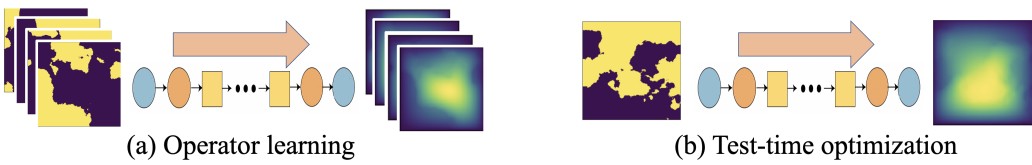

(a) Operator learning     (b) Test-time optimization

Figure 1: PINO: combine operator learning and PINN optimization.
(a) learn the solution operator from a family of equations. (b) use the learned operator as ansatz to solve for a specific instance.

**Operator learning and neural operators.**   A recent alternative approach is to learn the solution operator of a family of PDEs, defined by the map from the input–initial conditions and boundary conditions, to the output–solution functions. In this case, usually, a dataset of input-output pairs from an existing solver is given. There are two main aspects to consider (a) model: to design models for learning highly complicated PDE solution operators, and (b) data: to be data-efficient and to improve generalization. Recent advances in operator learning replace traditional convolutional neural networks and U-Nets from computer vision with operator-based model tailored to PDEs with greatly improved model expressiveness (Li et al., 2020c; Lu et al., 2019; Patel et al., 2021; Wang et al., 2020a; Duvall et al., 2021). Specifically, the neural operator generalizes the neural network to the operator setting where the input and output spaces are infinite-dimensional. The framework has shown success in learning resolution-invariant solution operators for highly non-linear problems such as turbulence flow (Li et al., 2020b;a). However, the data challenges remain: (1) the need for training data, which assumes an existing solver or experimental setup, (2) the non-negligible generalization error, and (3) extrapolation to unseen conditions. These issues can be addressed by adding physics or PDE constraints to operator learning (Zhu et al., 2019; Wang et al., 2021b; Zhang et al., 2021).

**Our contributions.**   To overcome the shortcomings of both physics-informed optimization and data-driven operator learning, we propose the physics-informed neural operator (PINO) that combines operator learning with equation solving (test-time optimization). It requires fewer or no data points to learn the operator and generalizes better. In PINO, we use the pre-trained operator as the ansatz to optimize for the solution function at test time, which reduces the generalization error. Compared to PINN, PINO has a much better optimization landscape and representation space, and hence, PINO converges faster and more accurately. Our contributions can be summarized as follows:

- We propose the physics-informed neural operator (PINO), combining the operator-learning and physics-informed settings. We introduce the pre-training and test-time optimization schemes that utilize both the data and equation constraints (whichever are available). We develop an efficient method to compute the exact gradient for neural operators to incorporate the equation constraints.

- By utilizing pre-trained operator ansatz, PINO overcomes the challenge of propagating information from the initial condition to future time steps with (soft) physics constraints. It can solve the 2d transient flow over an extremely long time period, where PINN and DeepONet (Lu et al., 2019) fail. Even without any pre-training and using only PDE constraints for the given instance, PINO still outperforms PINN by 20x smaller error and 25x speedup on the chaotic Kolmogorov flow, demonstrating superior expressivity of the neural operator over standard neural networks.

- By utilizing the equation constraints, PINO requires fewer or no training data and generalizes better compared to FNO (Li et al., 2020c). On average it has 7% smaller error on the transient and Kolmogorov flows, while matching the speedup of FNO (400x) compared to the GPU-based pseudo-spectral solver (He & Sun, 2007), matching FNO. Further, the pre-trained PINO model on the Navier Stokes equation can be easily transferred to different Reynolds numbers ranging from 100 to 500 using test-time optimization.

- We propose the forward and backward PINO models for inverse problems. Our approach accurately recovers the coefficient function in the Darcy flow which is 3000x faster than the conventional solvers using accelerated MCMC (Cotter et al., 2013).

Our major novelty and contributions are to use the pre-trained operator ansatz with instance-wise fine-tuning to overcome the optimization challenges in PINN and the generalization challenges

in operator learning. Previous works such as PINN-DeepONet (Wang et al., 2021b) and Physics-constrained modeling (Zhu et al., 2019) use the PDE constraints in operator learning, like we do during the pre-training phase in PINO. However, we propose several methodological advances as well as extensive experiments to understand the optimization and generalization challenges. Our methodological advances include: (1) Instance-wise fine-tuning at test-time to further improve the fidelity of the operator ansatz. (2) Efficient Fourier-space methods for computing derivatives present in the PDE loss. (3) Efficient learning through the design of data augmentation and loss functions. (4) Novel formulation for inverse problems that results in accurate recovery as well as good speedups.

**PINN vs. PINO: pointwise vs. function-wise optimization.** The neural operator ansatz in PINO has an easier optimization landscape and a more expressive representation space compared to the neural networks ansatz in PINN. The neural operator parameterizes the solution function as an aggregation of basis functions, and hence, the optimization is in the function space. This is easier than just optimizing a single function as in PINN. Further, we can learn these basis functions in the pre-training phase which makes the test-time optimization on the querying instance even easier. In this case, PINO does not need to solve from scratch. It just fine-tunes the solution function parameterized by the solution operator. Thus, PINO is much faster and more accurate compared to PINN.

## 2 PRELIMINARIES AND PROBLEM SETTINGS

### 2.1 PROBLEM SETTINGS

We consider two natural class of PDEs. In the first, we consider the stationary system

$$
\begin{aligned}
\mathcal{P}(u, a) = 0, && \text{in } D \subset \mathbb{R}^d \\
u = g, && \text{in } \partial D
\end{aligned}
\tag{1}
$$

where $D$ is a bounded domain, $a \in \mathcal{A} \subseteq \mathcal{V}$ is a PDE coefficient/parameter, $u \in \mathcal{U}$ is the unknown, and $\mathcal{P} : \mathcal{U} \times \mathcal{A} \to \mathcal{F}$ is a possibly non-linear partial differential operator with $(\mathcal{U}, \mathcal{V}, \mathcal{F})$ a triplet of Banach spaces. Usually the function $g$ is a fixed boundary condition (potentially can be entered as a parameter). This formulation gives rise to the solution operator $\mathcal{G}^\dagger : \mathcal{A} \to \mathcal{U}$ defined by $a \mapsto u$. A prototypical example is the second-order elliptic equation $\mathcal{P}(u, a) = -\nabla \cdot (a \nabla u) + f$.

In the second setting, we consider the dynamical system

$$
\begin{aligned}
\frac{du}{dt} = \mathcal{R}(u), && \text{in } D \times (0, \infty) \\
u = g, && \text{in } \partial D \times (0, \infty) \\
u = a && \text{in } \bar{D} \times \{0\}
\end{aligned}
\tag{2}
$$

where $a = u(0) \in \mathcal{A} \subseteq \mathcal{V}$ is the initial condition, $u(t) \in \mathcal{U}$ for $t > 0$ is the unknown, and $\mathcal{R}$ is a possibly non-linear partial differential operator with $\mathcal{U}$, and $\mathcal{V}$ Banach spaces. As before, we take $g$ to be a known boundary condition. We assume that $u$ exists and is bounded for all time and for every $u_0 \in \mathcal{U}$. This formulation gives rise to the solution operator $\mathcal{G}^\dagger : \mathcal{A} \to C\big((0, T]; \mathcal{U}\big)$ defined by $a \mapsto u$. Prototypical examples include the Burgers' equation and the Navier-Stokes equation.

### 2.2 SOLVING EQUATION USING THE PHYSICS-INFORMED LOSS (PINN)

Given an instance $a$ and a solution operator $\mathcal{G}^\dagger$ defined by equations (1) or (2) , we denote by $u^\dagger = \mathcal{G}^\dagger(a)$ the unique ground truth. The equation solving task is to approximate $u^\dagger$. This setting consists of the ML-enhanced conventional solvers such as learned finite element, finite difference, and multigrid solvers (Kochkov et al., 2021; Pathak et al., 2021; Greenfeld et al., 2019), as well as purely neural network-based solvers such as the Physics-Informed Neural Networks (PINNs), Deep Galerkin Method, and Deep Ritz Method (Raissi et al., 2019; Sirignano & Spiliopoulos, 2018; Weinan & Yu, 2018). Especially, these PINN-type methods use a neural network $u_\theta$ with parameters $\theta$ as the the ansatz to approximate the solution function $u^\dagger$ . The parameters $\theta$ are found by minimizing the physics-informed loss with exact derivatives computed using automatic-differentiation (autograd). In the stationary case, the physics-informed loss is defined by minimizing the l.h.s. of equation (1) in

the squared norm of $\mathcal{F}$. A typical choice is $\mathcal{F} = L^2(D)$, giving the loss function

$$
\begin{aligned}
\mathcal{L}_{\text{pde}}(a, u_\theta) &= \|\mathcal{P}(a, u_\theta)\|_{L^2(D)}^2 + \alpha\|u_\theta|_{\partial D} - g\|_{L^2(\partial D)}^2 \\
&= \int_D |\mathcal{P}(u_\theta(x), a(x))|^2 \mathrm{d}x + \alpha \int_{\partial D} |u_\theta(x) - g(x)|^2 \mathrm{d}x
\end{aligned}
\tag{3}
$$

In the case of a dynamical system, it minimizes the residual of equation (2) in some natural norm up to a fixed final time $T > 0$. A typical choice is the $L^2\big((0, T]; L^2(D)\big)$ norm, yielding

$$
\begin{aligned}
\mathcal{L}_{\text{pde}}(a, u_\theta) &= \int_0^T \int_D |\frac{du_\theta}{dt}(t, x) - \mathcal{R}(u_\theta)(t, x)|^2 \mathrm{d}x\mathrm{d}t + \alpha \int_0^T \int_{\partial D} |u_\theta(t, x) - g(t, x)|^2 \mathrm{d}x\mathrm{d}t \\
&+ \beta \int_D |u_\theta(0, x) - a(x)|^2 \mathrm{d}x
\end{aligned}
\tag{4}
$$

The PDE loss consists of the physics loss in the interior and the data loss on the boundary and initial conditions, with hyper-parameters $\alpha, \beta > 0$. It can be generalized to variational form as in (Weinan & Yu, 2018).

**Challenges of PINN**  PINNs take advantage of the universal approximability of neural networks, but, in return, suffer from the low-frequency induced bias. Empirically, PINNs often fail to solve challenging PDEs when the solution exhibits high-frequency or multi-scale structure (Wang et al., 2021a; 2020b; Fuks & Tchelepi, 2020; Raissi et al., 2020). Further, as an iterative solver, PINNs have difficulty propagating information from the initial condition or boundary condition to unseen parts of the interior or to future times (Dwivedi & Srinivasan, 2020). For example, in challenging problems such as turbulence, PINNs are only able to solve the PDE on a relatively small domain (Jin et al., 2021), or otherwise, require extra observational data which is not always available in practice (Raissi et al., 2020; Cai et al., 2021). In this work, we propose to overcome the challenges posed by the optimization by integrating operator learning with PINNs.

## 2.3  LEARNING THE SOLUTION OPERATOR (NEURAL OPERATOR)

An alternative setting is to learn the solution operator $\mathcal{G}$. Given a PDE as defined in (1) or (2) and the corresponding solution operator $\mathcal{G}^\dagger$, one can use a neural operator $\mathcal{G}_\theta$ with parameters $\theta$ as a surrogate model to approximate $\mathcal{G}^\dagger$. Usually we assume a dataset $\{a_j, u_j\}_{j=1}^N$ is available, where $\mathcal{G}^\dagger(a_j) = u_j$ and $a_j \sim \mu$ are i.i.d. samples from some distribution $\mu$ supported on $\mathcal{A}$. In this case, one can optimize the solution operator by minimizing the empirical data loss on a given data pair

$$
\mathcal{L}_{\text{data}}(u, \mathcal{G}_\theta(a)) = \|u - \mathcal{G}_\theta(a)\|_{\mathcal{U}}^2 = \int_D |u(x) - \mathcal{G}_\theta(a)(x)|^2 \mathrm{d}x
\tag{5}
$$

where we assume the setting of (1) for simplicity of the exposition. The operator data loss is defined as the average error across all possible inputs

$$
\mathcal{J}_{\text{data}}(\mathcal{G}_\theta) = \|\mathcal{G}^\dagger - \mathcal{G}_\theta\|_{L_\mu^2(\mathcal{A};\mathcal{U})}^2 = \mathbb{E}_{a\sim\mu}[\mathcal{L}_{\text{data}}(a, \theta)] \approx \frac{1}{N} \sum_{j=1}^N \int_D |u_j(x) - \mathcal{G}_\theta(a_j)(x)|^2 \mathrm{d}x.
\tag{6}
$$

Similarly, one can define the operator PDE loss as

$$
\mathcal{J}_{\text{pde}}(\mathcal{G}_\theta) = \mathbb{E}_{a\sim\mu}[\mathcal{L}_{\text{pde}}(a, \mathcal{G}_\theta(a))].
\tag{7}
$$

In general, it is non-trivial to compute the derivatives $d\mathcal{G}_\theta(a)/dx$ and $d\mathcal{G}_\theta(a)/dt$ for model $\mathcal{G}_\theta$. In the following section, we will discuss how to compute these derivatives for Fourier neural operator.

## 2.4  NEURAL OPERATORS

In this work, we will focus on the neural operator model designed for the operator learning problem. The neural operator, proposed in (Li et al., 2020c), is formulated as an generalization of standard deep neural networks to operator setting. Neural operator compose linear integral operator $\mathcal{K}$ with pointwise non-linear activation function $\sigma$ to approximate highly non-linear operators.

**Definition 1 (Neural operator $\mathcal{G}_\theta$)** *Define the neural operator*

$$\mathcal{G}_\theta := \mathcal{Q} \circ (W_L + \mathcal{K}_L) \circ \cdots \circ \sigma(W_1 + \mathcal{K}_1) \circ \mathcal{P} \tag{8}$$

*where $\mathcal{P} : \mathbb{R}^{d_a} \to \mathbb{R}^{d_1}$, $\mathcal{Q} : \mathbb{R}^{d_L} \to \mathbb{R}^{d_u}$ are the pointwise neural networks that encode the lower dimension function into higher dimensional space and vice versa. The model stack $L$ layers of $\sigma(W_l + \mathcal{K}_l)$ where $W_l \in \mathbb{R}^{d_{l+1} \times d_l}$ are pointwise linear operators (matrices), $\mathcal{K}_l : \{D \to \mathbb{R}^{d_l}\} \to \{D \to \mathbb{R}^{d_{l+1}}\}$ are integral kernel operators, and $\sigma$ are fixed activation functions. The parameters $\theta$ consists of all the parameters in $\mathcal{P}, \mathcal{Q}, W_l, \mathcal{K}_l$.*

Recently, Li et al. (2020a) proposes the Fourier neural operator (FNO) that deploys convolution operator for $\mathcal{K}$. In this case, it can apply the Fast Fourier Transform (FFT) to efficiently compute $\mathcal{K}$. This leads to a fast architecture that obtains state-of-the-art results for PDE problems.

**Definition 2 (Fourier convolution operator $\mathcal{K}$)** *Define the Fourier convolution operator*

$$\big(\mathcal{K}v_t\big)(x) = \mathcal{F}^{-1}\Big(R \cdot (\mathcal{F}v_t)\Big)(x) \qquad \forall x \in D \tag{9}$$

*where $R$ is part of the parameter $\theta$ to be learn.*

**Challenges of Operator learning.** Operator learning is similar to the supervised learning in computer vision and language where data play a very important role. One needs to assume the training points and testing points follow the same problem setting and the same distribution. Especially, the previous FNO model trained on one coefficient (e.g. Reynolds number) or one geometry cannot be easily generalized to another. Moreover, for more challenging PDEs where the solver is very slow or the solver is even not existent, it is hard to gather a representative dataset. On the other hand, since FNO doesn't use any knowledge of the equation, it is cannot get arbitrary close to the ground truth by using higher resolution as in conventional solvers, leaving a gap of generalization error. These challenges limit FNO's applications beyond accelerating the solver. In the following section, we will introduce the PINO framework to overcome these problems by using the equation constraints.

## 3 PHYSICS-INFORMED NEURAL OPERATOR (PINO)

We propose the PINO framework that uses one neural operator model $\mathcal{G}_\theta$ for solving both operator learning problems and equation solving problems. It consists of two phases

- **Pre-train the solution operator**: learn a neural operator $\mathcal{G}_\theta$ to approximate $\mathcal{G}^\dagger$ using either/both the data loss $\mathcal{J}_{data}$ and/or the PDE loss $\mathcal{J}_{pde}$ which can be additional to the dataset.

- **Test-time optimization**: use $\mathcal{G}_\theta(a)$ as the ansatz to approximate $u^\dagger$ with the pde loss $\mathcal{L}_{pde}$ and/or an additional operator loss $\mathcal{L}_{op}$ obtained from the pre-train phase.

### 3.1 PRE-TRAIN: OPERATOR LEARNING WITH PDE LOSS

Given a fixed amount of data $\{a_j, u_j\}$, the data loss $\mathcal{J}_{data}$ offers a stronger constraint compared to the PDE loss $\mathcal{J}_{pde}$. However, the PDE loss does not require a pre-existing dataset. One can sample virtual PDE instances by drawing additional initial conditions or coefficient conditions $a_j \sim \mu$ for training, as shown in Algorithm 1. In this sense, we have access to the unlimited dataset by sampling new $a_j$ in each iteration.

### 3.2 TEST-TIME OPTIMIZATION: SOLVING EQUATION WITH OPERATOR ANSATZ

Given a learned operator $\mathcal{G}_\theta$, we use $\mathcal{G}_\theta(a)$ as the ansatz to solve for $u^\dagger$. The optimization procedure is similar to PINNs where it computes the PDE loss $\mathcal{L}_{pde}$ on $a$, except that we propose to use a neural operator instead of a neural network. Since the PDE loss is a soft constraints and challenging to optimize, we also add an optional operator loss $\mathcal{L}_{op}$ (anchor loss) to bound the further optimized model from the pre-trained model

$$\mathcal{L}_{op}\big(\mathcal{G}_{\theta_i}(a), \mathcal{G}_{\theta_0}(a)\big) := \|\mathcal{G}_{\theta_i}(a) - \mathcal{G}_{\theta_0}(a)\|_{\mathcal{U}}^2$$

---

**Algorithm 1:** Pre-training scheme for Physics-informed neural operator learning

---

**Data:** Data: input output function pairs $\{a_j, u_j\}_{j=1}^N$
**Result:** Neural operator $\mathcal{G} : \mathcal{A} \to \mathcal{U}$.

1 **for** $i = 0, 1, 2 \ldots$ **do**
2      Compute $\mathcal{L}_{data}$ and $\mathcal{L}_{pde}$ on $(a_i, u_i)$. Update neural operator $\mathcal{G}$;
3      **for** $j = 1$ **to** $K$ **do**
4          Sample $a'$ from distribution $\mu$;
5          Compute $\mathcal{L}_{pde}$ on $a'$. Update neural operator $\mathcal{G}$;
6      **end**
7 **end**

---

where $\mathcal{G}_{\theta_i}(a)$ is the model at $i^{th}$ training epoch. We update the operator $\mathcal{G}_\theta$ using the loss $\mathcal{L}_{pde} + \alpha \mathcal{L}_{op}$. It is possible to further apply optimization techniques to fine-tune the last fewer layers of the neural operator and progressive training that gradually increase the grid resolution and use finer resolution in test time.

**Optimization landscape.** Using the operator as the ansatz has two major advantages: (1) PINN does point-wise optimization, while PINO does optimization in the space of functions. In the linear integral operation $\mathcal{K}$, the operator parameterizes the solution function as a sum of the basis function. Optimization of the set of coefficients and basis is easier than just optimizing a single function as in PINNs. (2) we can learn these basis functions in the operator learning phase which makes the later test-time optimization even easier. In PINO, we do not need to propagate the information from the initial condition and boundary condition to the interior. It just requires fine-tuning the solution function parameterized by the solution operator.

**Trade-off** (1) complexity and accuracy: the test-time optimization is an option to spend more computation in exchange of better accuracy. When the speed is most desired, one should use the pre-trained model to directly output the prediction. However, if one wants to solve for a specific instance accurately, then the person can use test-time optimization to correct the generalization error posed in operator learning. (2) resolution effects on optimization landscape and truncation error: using higher resolution and finer grid will reduce the truncation error. However, it may make the optimization instable. Using hard constraints such as the anchor loss $\mathcal{L}_{op}$ relieve such problem.

### 3.3 DERIVATIVE OF NEURAL OPERATORS

In order to use the equation loss $\mathcal{L}_{pde}$, one of the major technical challenge is to efficiently compute the derivatives $D(\mathcal{G}_\theta a) = d(\mathcal{G}_\theta a)(x)/dx$ for neural operators. In this section, we discuss three efficient methods to compute the derivatives of neural operator $\mathcal{G}_\theta$ as defined in 8.

**Numerical differentiation.** A simple but efficient approach is to use conventional numerical gradients such as finite difference and Fourier gradient (Zhu et al., 2019; Gao et al., 2021). These numerical gradient are fast and memory-efficient: given a $n$-points grid, finite difference requires $O(n)$ and Fourier method require $O(n \log n)$. However, they face the same challenges as the corresponding numerical solvers: finite difference methods require a fine-resolution uniform grid; spectral methods require smoothness and uniform grids. Especially. These numerical errors on the gradient will be amplified on the output solution.

**Autograd.** Similar to PINN (Raissi et al., 2019), the most general way to compute the exact gradient is to use the auto-differentiation library of neural networks (autograd). To apply autograd, one needs to use a neural network to parameterize the solution function $\hat{u} : x \mapsto u(x)$. However, it is not straightforward to write out the solution function in the neural operator which directly outputs the numerical solution $u = \mathcal{G}_\theta(a)$ on a grid, especially for FNO which uses FFT. To apply autograd, we design a query function $\hat{u}$ that input $x$ and output $u(x)$. Recall $\mathcal{G}_\theta := \mathcal{Q} \circ (W_L + \mathcal{K}_L) \circ \cdots \circ \sigma(W_1 + \mathcal{K}_1) \circ \mathcal{P}$ and $u = \mathcal{G}_\theta a = \mathcal{Q} v_L = \mathcal{Q}(W_L + \mathcal{K}_L) v_{L-1} \ldots$. Since $\mathcal{Q}$ is pointwise,

$$u(x) = \mathcal{Q}(v_L(x)) = \mathcal{Q}((W_L v_{L-1})(x) + \mathcal{K}_L v_{L-1}(x)) \tag{10}$$

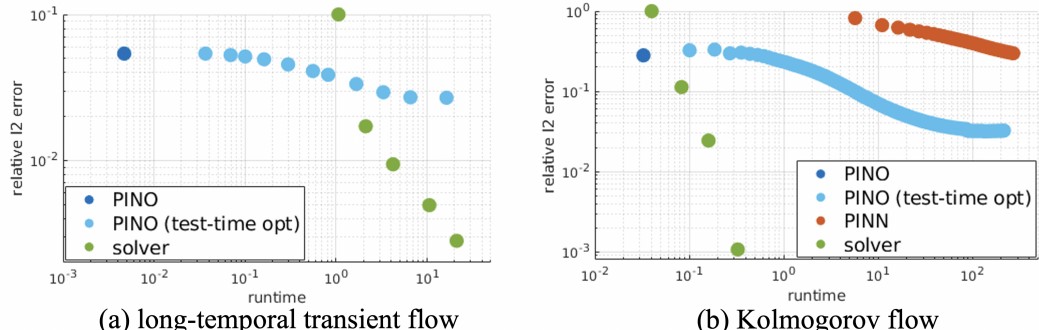

(a) long-temporal transient flow          (b) Kolmogorov flow

(a) the long-temporal transient flow with $Re \sim 20, T = 50$. PINO can output the full trajectory in one step, which leads to 400x speedup compared to the solver. PINN cannot converge to a reasonable error rate due to the long time window. (b) the chaotic Kolmogorov flow with $Re =\sim 500, T = 0.5$. PINO retains 10x speed up compared to the solver, while converges much faster compared to PINN.

Figure 2: The accuracy-complexity trade-off on PINO, PINN, and the pseudo-spectral solver.

To query any location $x$, we need to evaluate the Fourier convolution $\mathcal{K}_L v_{L-1}(x)$ and the bias part $(W_L v_{L-1})(x)$. For the first part, we can directly query the Fourier coefficient $R \cdot \mathcal{F} v_t$ at spatial location $x$ without doing the inverse FFT $\mathcal{F}$. For the second part, we can define the query function as an interpolation or a low-rank integral operator as in (Kovachki et al., 2021). Then we can use autograd to take the exact derivative for (10). The autograd method is general and exact, however, it is less efficient. Since the number of parameters $|\theta|$ is usually much greater than the grid size $n$, the numerical methods are indeed significantly faster. Empirically, the autograd method is usually slower and memory-consuming.

**Exact gradients.** We develop an efficient and exact gradient method based on the architecture of the neural operator. The idea is similar to the autograd, but we explicitly write out the gradient on the Fourier space and apply the chain rule. Given the explicit form (10), $D(\mathcal{K}_L v_{L-1})$ can be directly compute on the Fourier space. The linear part $D(W_L v_{L-1})$ can be interpolated using the Fourier method. Therefore to exactly compute the derivative of $v_L$, one just needs to run the numerical Fourier gradient. Then we just need to apply chain rule for Jacobian $Ju = J(\mathcal{Q} \circ v_L) = J(\mathcal{Q}(v_L)) \circ J(v_L)$ and Hessian $Hu = H(\mathcal{Q} \circ v_L) = J(v_L) \circ H(\mathcal{Q}(v_L)) \circ J(v_L) + J(\mathcal{Q}(v_L)) \circ H(v_L)$. Since $\mathcal{Q}$ is pointwise, this can be much simplified. In the experiments, we mostly use the exact gradient and the numerical Fourier gradient.

## 4 EXPERIMENTS

In this section, we conduct empirical experiments to examine the efficacy of the proposed PINO. In 4.1, we show the physics constraints helps operator-learning with fewer data and better generalization. Then in 4.2, we investigate how PINO uses operator ansatz to solve harder equations with improved speed and accuracy. We study on three concrete cases of PDEs on Burgers' Equation, Darcy Equation, and Navier-Stokes equation.

**Burgers' Equation.** The 1-d Burgers' equation is a non-linear PDE with periodic boundary conditions where $u_0 \in L^2_{\text{per}}((0,1); \mathbb{R})$ is the initial condition and $\nu = 0.01$ is the viscosity coefficient. We aim to learn the operator mapping the initial condition to the solution, $\mathcal{G}^\dagger : u_0 \mapsto u|_{[0,1]}$.

$$\partial_t u(x,t) + \partial_x(u^2(x,t)/2) = \nu \partial_{xx} u(x,t), \qquad x \in (0,1), t \in (0,1]$$
$$u(x,0) = u_0(x), \qquad x \in (0,1) \tag{11}$$

**Darcy Flow.** The 2-d steady-state Darcy Flow equation on the unit box which is the second order linear elliptic PDE with a Dirichlet boundary where $a \in L^\infty((0,1)^2; \mathbb{R}_+)$ is a piecewise constant diffusion coefficient and $f = 1$ is a fixed forcing function. We are interested in learning the operator

mapping the diffusion coefficient to the solution, $\mathcal{G}^\dagger : a \mapsto u$. Note that although the PDE is linear, the operator $\mathcal{G}^\dagger$ is not.

$$
\begin{aligned}
-\nabla \cdot (a(x)\nabla u(x)) &= f(x) & x \in (0,1)^2 \\
u(x) &= 0 & x \in \partial(0,1)^2
\end{aligned}
\tag{12}
$$

**Navier-Stokes Equation.** We consider the 2-d Navier-Stokes equation for a viscous, incompressible fluid in vorticity form on the unit torus, where $u \in C([0,T]; H_{\mathrm{per}}^r((0,l)^2; \mathbb{R}^2))$ for any $r > 0$ is the velocity field, $w = \nabla \times u$ is the vorticity, $w_0 \in L_{\mathrm{per}}^2((0,l)^2; \mathbb{R})$ is the initial vorticity, $\nu \in \mathbb{R}_+$ is the viscosity coefficient, and $f \in L_{\mathrm{per}}^2((0,l)^2; \mathbb{R})$ is the forcing function. We want to learn the operator mapping the vorticity from the initial condition to the full solution $\mathcal{G}^\dagger : w_0 \mapsto w|_{[0,T]}$.

$$
\begin{aligned}
\partial_t w(x,t) + u(x,t) \cdot \nabla w(x,t) &= \nu \Delta w(x,t) + f(x), & x \in (0,l)^2, t \in (0,T] \\
\nabla \cdot u(x,t) &= 0, & x \in (0,l)^2, t \in [0,T] \\
w(x,0) &= w_0(x), & x \in (0,l)^2
\end{aligned}
\tag{13}
$$

Specially, we consider two problem settings:

- **Long temporal transient flow**: we study the build-up of the flow from the initial condition $u_0$ near-zero velocity to $u_T$ that reaches the ergodic state. We choose $t \in [0,50]$, $l = 1$, $Re = 20$ as in Li et al. (2020a). The main challenge is to predict the long time interval.

- **Chaotic Kolmogorov flow**: In this case $u$ lies in the attractor where arbitrary starting time $t_0$. We choose $t \in [t_0, t_0 + 0.5]$ or $[t_0, t_0 + 1]$, $l = 1$, $Re = 500$ similar to Li et al. (2021). The main challenge is to capture the small details that evolve chaotically.

### 4.1 OPERATOR LEARNING WITH PDE LOSSES

We first show the effectiveness of applying the physics constraints $\mathcal{J}_{pde}$ to learn the solution operator.

**Burgers equation and Darcy equation.** PINO can learn the solution operator without any data on simpler problems such as Burgers and Darcy. Compared to other PDE-constrained operators, PINO is more expressive and thereby achieves better accuracy. On Burgers (11), PINN-DeepONet achieves **1.38%** (Wang et al., 2021b); PINO achieves **0.37%**. Similarly, on Darcy flow (12), PINO outperforms FNO by utilizing physics constraints, as shown in Table 1. For these simpler equation, test-time optimization may not be needed. The implementation detail and and the search space of parameters are included in the Appendix A.1 and A.2.

### 4.2 SOLVE EQUATION USING OPERATOR ANSATZ

**Long temporal transient flow.** It is extremely challenging to propagate the information from the initial condition to future time steps over such a long interval $T = [0,50]$ just using the soft physics constraint. None of the PINN, DeepONet, and PINO (from scratch without pre-training) can handle this case (error $> 50\%$), no matter solving the full interval at once or solving per smaller steps. However, when the pre-training data is available for PINO, we can use the learned neural operator ansatz and the anchor loss $\mathcal{L}_{op}$. The anchor loss is a hard constraint that makes the optimization much easier. Providing $N = 4800$ training data, the PINO without test-time optimization achieves **2.87%** error, lower than FNO **3.04%** and it retains a 400x speedup compared to the GPU-based pseudo-spectral solver (He & Sun, 2007), matching FNO. Further doing test time optimization with the anchor loss and PDE loss, PINO reduces the error to **1.84%**.

**Chaotic Kolmogorov flow.** We show an empirical study on how PINO can improve the generalization of neural operators by enforcing more physics. For this experiment, the training set consists of 4000 data points of initial condition and corresponding solution. Using Algorithm 1, we can sample unlimited additional initial conditions from Gaussian random field at any resolution. Table 2 and 4 compare the generalization error of neural operators trained by different schemes and different amounts of simulated data. The result shows that training neural operator with additional PDE instances consistently improves the generalization error on all three resolutions we are evaluating.

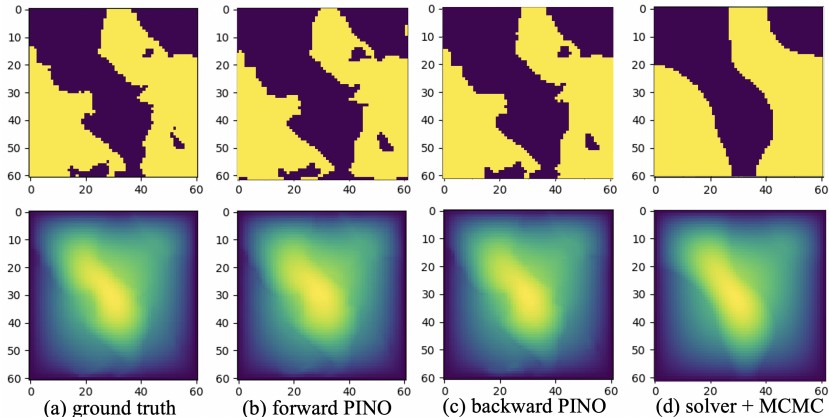

Figure 3: Darcy inverse problem

Based on the solution operators learned in the above operator-learning section, we continue to do test-time optimization. The results are shown in Figure and Table 3. Overall, PINO outperforms PINN by **20x** smaller error and **25x** speedup. Using pre-trained model make PINO converge faster. The implementation detail and the search space of parameters are included in the Appendix B.2.

**Transfer Reynolds numbers.** The extrapolation of different parameters and conditions is one of the biggest challenges for ML-based methods. By doing test-time optimization, the pre-trained PINO model on the Kolmogorov flow can be easily transferred to different Reynolds numbers ranging from 100 to 500 in test-time optimization as shown in Table 5. Such property envisions broad applications.

## 4.3 INVERSE PROBLEM

One of the major advantages of the physics-informed method is to solve the inverse problem. In this case, we investigate PINO on the inverse problem of the Darcy equation to recover the coefficient function $a^\dagger$ from the given solution function $u^\dagger$. We assume a dataset $\{a_j, u_j\}$ is available to pre-train the operator. We propose two formulations of PINO for the inverse problem:

- **Forward model**: Learn the forward operator $\mathcal{G}_\theta : a \mapsto u$ with data. Initialize $\hat{a}$ to approximate $a^\dagger$. Optimize $\hat{a}$ using
$$\mathcal{J}_{forward} := \mathcal{L}_{pde}(\hat{a}, u^\dagger) + \mathcal{L}_{data}(\mathcal{G}_\theta(\hat{a})) + R(\hat{a}).$$

- **Backward model**: Learn the backward operator $\mathcal{F}_\theta : u \mapsto a$ with data. Use $\mathcal{F}_\theta(u^\dagger)$ to approximate $a^\dagger$. Optimize $\mathcal{F}_\theta$ using
$$\mathcal{J}_{backward} := \mathcal{L}_{pde}(\mathcal{F}_\theta(u^\dagger), u^\dagger) + \mathcal{L}_{op}(\mathcal{F}_\theta(u^\dagger), \mathcal{F}_{\theta_0}(u^\dagger)) + R(\mathcal{F}_\theta(u^\dagger))$$

Where $R(a)$ is the regularization term. We use the PDE loss $\mathcal{L}_{pde}$ to deal with the small error in $\mathcal{G}_\theta$ and the ill-defining issue of $\mathcal{F}_\theta$.

We perform a showcase example on the Darcy equation, where we are given the full-field observation of $u$ (clean data) to recover the coefficient function $a$. The coefficient function $a$ is piecewise constant (representing two types of media), so the inverse problem can be viewed as a classification problem. We define $R(a)$ as the total variance. As shown in Figure 3, the PINO backward model has the best performance: it has **2.29%** relative l2 error on the output $u$ and **97.10%** classification accuracy on the input $a$; the forward model has 6.43% error on the output and 95.38% accuracy on the input. As a reference, we compare the PINO inverse frameworks with PINN and the conventional solvers using the accelerated MCMC method with 500,000 steps (Cotter et al., 2013). The posterior mean of the MCMC has 4.52% error and 90.30% respectively ( Notice the Bayesian method outputs the posterior distribution, which is beyond obtaining a maximum a posteriori estimation) Meanwhile, PINO methods are 3000x faster compared to MCMC. PINN does not converge in this case. The major advantage of the PINO backward model compared to the PINO forward model is that it uses a neural operator $\mathcal{F}_\theta(u^\dagger)$ as the ansatz for the coefficient function. Similar to the forward problem, the operator ansatz has an easier optimization landscape while being expressive.

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

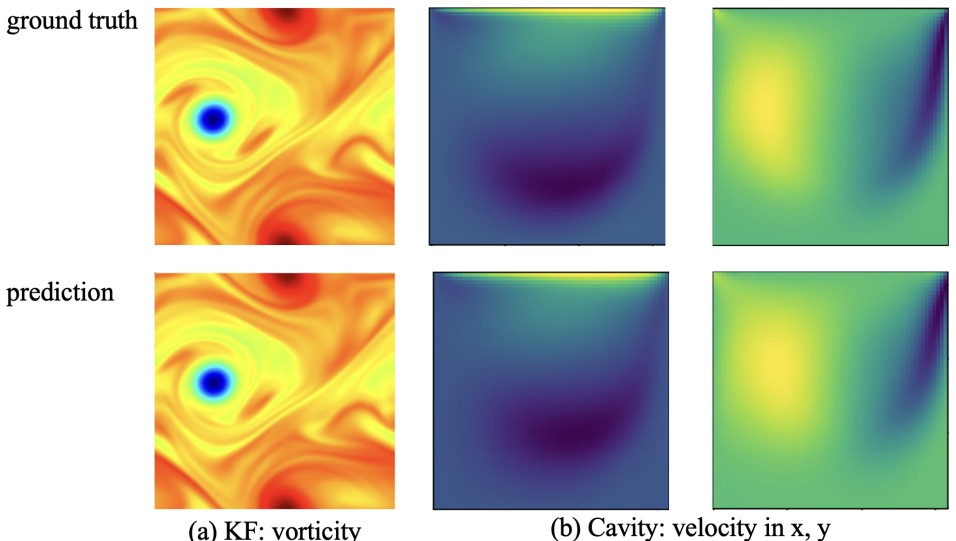

(a) KF: vorticity    (b) Cavity: velocity in x, y

Figure 4: PINO on Kolmogorov flow (left) and Lid-cavity flow (right)

## A  IMPLEMENTATION DETAILS

In this section, we list the detailed experiment setups and parameter searching for each experiment in Section 4. Without specification, we use Fourier neural operator backbone with $width = 64$, $mode = 8$, $L = 4$ and GeLU activation. The numerical experiments are performed on Nvidia V100 GPUs.

### A.1  BURGERS EQUATION

We use the 1000 initial conditions $u_0 \sim \mu$ where $\mu = \mathcal{N}(0, 625(-\Delta + 25I)^{-2})$ to train the solution operator on PINO with $width = 64$, $mode = 20$, and GeLU activation. We use the numerical method to take the gradient. We use Adam optimizer with the learning rate 0.001 that decays by half every 100 epochs. 500 epochs in total. The total training time is about $1250s$ on a single Nvidia 3090 GPU. PINO achieves **0.37%** relative l2 error averaged over 200 testing instances. PINN-DeepONet achieves **1.38%** which is taken from Wang et al. (2021b) which uses the same problem setting.

### A.2  DARCY FLOW

We use the 1000 coefficient conditions $a$ to train the solution operator where $a \sim \mu$ where $\mu = \psi_{\#}\mathcal{N}(0, (-\Delta + 9I)^{-2})$, $\psi(a(x)) = 12$ if $a(x) \geq 0$; $\psi(a(x)) = 3$ if $a(x) < 0$. The zero boundary condition is enforced by multiplying a mollifier $m(x) = \sin(\pi x)\sin(\pi y)$ for all methods. The parameter of PINO on Darcy Flow is the same as in the Burgers equation above. Regarding the implementation detail of the baselines: as for FNO, we use the same hyperparameters as its paper did (Li et al., 2020c); DeepONet (Lu et al., 2019) did not study Darcy flow so we grid search the hyperparameters of DeepONet: depth from 2 to 12, width from 50 to 100. The best result of DeepONet is achieved by depth 8, width 50. The results are shown in Table 1.

| Method | Solution error | Equation error |
|---|---|---|
| DeepONet with data (Lu et al., 2019) | $6.97 \pm 0.09\%$ | - |
| FNO with data (Li et al., 2020c) | $1.98 \pm 0.05\%$ | $1.3645 \pm 0.014$ |
| PINO with data | $1.22 \pm 0.03\%$ | $0.5740 \pm 0.008$ |
| PINO w/o data | $1.50 \pm 0.03\%$ | $0.4868 \pm 0.007$ |

Table 1: Operator learning on Darcy Flow.

A.3 LONG TEMPORAL TRANSIENT FLOW.

We study the build-up of the flow from the initial condition $u_0$ near-zero velocity to $u_T$ that reaches the ergodic state. We choose $T = 50, l = 1$ as in Li et al. (2020a). We choose the weight parameters of error $\alpha = \beta = 5$. The initial condition $w_0(x)$ is generated according to $w_0 \sim \mu$ where $\mu = \mathcal{N}(0, 7^{3/2}(-\Delta + 49I)^{-2.5})$ with periodic boundary conditions. The forcing is kept fixed $f(x) = 0.1(\sin(2\pi(x_1 + x_2)) + \cos(2\pi(x_1 + x_2)))$. We compare FNO, PINO (no test-time optimization), and PINO (with test-time optimization). They get $3.04\%$, $2.87\%$, and $1.84\%$ relative l2 error on the last time step $u(50)$ over 5 testing instance.

| # data samples | # additional PDE instances | Solution error | Equation error |
|---|---|---|---|
| 0 | 100k | 74.36% | 0.3741 |
| 0.4k | 0 | 33.32% | 1.8779 |
| 0.4k | 40k | 31.74% | 1.8179 |
| 0.4k | 160k | **31.32%** | 1.7840 |
| 4k | 0 | 25.15% | 1.8223 |
| 4k | 100k | **24.15%** | 1.6112 |
| 4k | 400k | **24.22%** | 1.4596 |

Table 2: Operator-learning on Kolmogorov flow $Re = 500$. Each relative $L_2$ test error is averaged over 300 instances, which is evaluated with resolution $128 \times 128 \times 65$. Complete results of other resolutions are reported in Table 4 in appendix.

A.4 CHAOTIC KOLMOGOROV FLOW.

For this experiment, $u$ lies in the attractor. We choose $T = 0.5$ or 1, and $l = 1$, similar to Li et al. (2021). The training set consists of 4000 initial condition functions and corresponding solution functions with a spatial resolution of $64 \times 64$ and a temporal resolution of 65. Extra initial conditions are generated from Gaussian random field $\mathcal{N}(0, 7^{3/2}(-\Delta + 49I)^{-5/2})$. We estimate the generalization error of the operator on a test set that contains 300 instances of Kolmogorov flow and reports the averaged relative $L_2$ error. Each neural operator is trained with 4k data points plus a number of extra sampled initial conditions. The Reynolds number in this problem is 500. The reported generalization error is averaged over 300 instances.

**Comparison study.** The baseline method PINN is implemented using library DeepXDE (Lu et al., 2021c) with TensorFlow as backend. We use the two-step optimization strategy (Adam (Kingma & Ba, 2014) and L-BFGS) following the same practice as NSFNet (Jin et al., 2021), which applies PINNs to solving Navier Stokes equations. We grid search the hyperparameters: network depth from 4 to 6, width from 50 to 100, learning rate from 0.01 to 0.0001, the weight of boundary loss from 10 to 100 for all experiments of PINNs. Comparison between PINO and PINNs on test-time optimization. The results are averaged over 20 instances of the Navier Stokes equation with Reynolds number 500. The best result is obtained by PINO using pre-trained operator ansatz and virtual sampling. The neural operator ansatz used here is trained over a set of 400 data points. The authors acknowledge that there could exist more sophisticated variants of PINN that performs better in our test cases.

| Method | # data samples | # additional PDE instances | Solution error ($w$) | Time cost |
|---|---|---|---|---|
| PINNs | - | - | 18.7% | 4577s |
| PINO | 0 | 0 | **0.9%** | 608s |
| PINO | 4k | 0 | **0.9%** | 536s |
| PINO | 4k | 160k | **0.9%** | **473s** |

Table 3: Equation-solving on Kolmogorov flow $Re = 500$.

A.5 TRANSFER LEARNING ACROSS REYNOLDS NUMBERS

We study the test-time optimization with different Reynolds numbers on the 1s Kolmogorov flow. For the higher Reynolds number problem $Re = 500, 400$, the pre-training operator shows better

| # data samples | # additional PDE instances | Resolution | Solution error | Equation error |
|---|---|---|---|---|
| 400 | 0 | 128x128x65 | 33.32% | 1.8779 |
| | | $64 \times 64 \times 65$ | 33.31% | 1.8830 |
| | | $32 \times 32 \times 33$ | 30.61% | 1.8421 |
| 400 | 40k | 128x128x65 | 31.74% | 1.8179 |
| | | $64 \times 64 \times 65$ | 31.72% | 1.8227 |
| | | $32 \times 32 \times 33$ | 29.60% | 1.8296 |
| 400 | 160k | 128x128x65 | **31.32%** | 1.7840 |
| | | $64 \times 64 \times 65$ | **31.29%** | 1.7864 |
| | | $32 \times 32 \times 33$ | **29.28%** | 1.8524 |
| 4k | 0 | $128 \times 128 \times 65$ | 0.2515 | 1.8223 |
| | | $64 \times 64 \times 65$ | 0.2516 | 1.8257 |
| | | $32 \times 32 \times 33$ | 0.2141 | 1.8468 |
| 4k | 100k | $128 \times 128 \times 65$ | 0.2415 | 1.6112 |
| | | $64 \times 64 \times 65$ | 0.2411 | 1.6159 |
| | | $32 \times 32 \times 33$ | 0.2085 | 1.8251 |
| 4k | 400k | $128 \times 128 \times 65$ | 0.2422 | 1.4596 |
| | | $64 \times 64 \times 65$ | 0.2395 | 1.4656 |
| | | $32 \times 32 \times 33$ | 0.2010 | 1.9146 |
| 0 | 100k | $128 \times 128 \times 65$ | 0.7436 | 0.3741 |
| | | $64 \times 64 \times 65$ | 0.7438 | 0.3899 |
| | | $32 \times 32 \times 33$ | 0.7414 | 0.5226 |

Table 4: Each neural operator is trained with 4k data points additionally sampled free initial conditions. The Reynolds number is 500. The reported generalization error is averaged over 300 instances. Training on additional initial conditions boosts the generalization ability of the operator.

convergence accuracy. In all cases, the pre-training operator shows better convergence speed as demonstrated in Figure 5. The results are shown in Table 5 where the error is averaged over 40 instances. Each row is a testing case and each column is a pre-trained operator.

| Testing RE | From scratch | 100 | 200 | 250 | 300 | 350 | 400 | 500 |
|---|---|---|---|---|---|---|---|---|
| 500 | 0.0493 | 0.0383 | 0.0393 | 0.0315 | 0.0477 | 0.0446 | 0.0434 | 0.0436 |
| 400 | 0.0296 | 0.0243 | 0.0245 | 0.0244 | 0.0300 | 0.0271 | 0.0273 | 0.0240 |
| 350 | 0.0192 | 0.0210 | 0.0211 | 0.0213 | 0.0233 | 0.0222 | 0.0222 | 0.0212 |
| 300 | 0.0168 | 0.0161 | 0.0164 | 0.0151 | 0.0177 | 0.0173 | 0.0170 | 0.0160 |
| 250 | 0.0151 | 0.0150 | 0.0153 | 0.0151 | 0.016 | 0.0156 | 0.0160 | 0.0151 |
| 200 | 0.00921 | 0.00913 | 0.00921 | 0.00915 | 0.00985 | 0.00945 | 0.00923 | 0.00892 |
| 100 | 0.00234 | 0.00235 | 0.00236 | 0.00235 | 0.00239 | 0.00239 | 0.00237 | 0.00237 |

Table 5: Reynolds number transfer learning. Each row is a test set of PDEs with corresponding Reynolds number. Each column represents the operator ansatz we use as the starting point of test-time optimization. For example, column header '100' means the operator ansatz is trained over a set of PDEs with Reynolds number 100. The relative $L_2$ errors is averaged over 40 instances of the corresponding test set.

# B ADDITIONAL EXPERIMENTS

## B.1 ADDITIONAL BASELINES

We add a comparison experiment against the Locally adaptive activation functions for PINN (LAAF-PINN) (Jagtap et al., 2020) and Self-Adaptive PINN (SA-PINN) (McClenny & Braga-Neto, 2020). For the Kolmogorov flow problem, we set Re=500, T=[0, 0.5]. We search among the following hyperparameters combinations: LAAF-PINN: n: 10, 100, learning rate: 0.1, 0.01, 0.001, depth 4, 6. SA-PINNs: learning rate 0.001, 0.005, 0.01, 0.05, network width 50, 100, 200, depth 4, 6, 8.

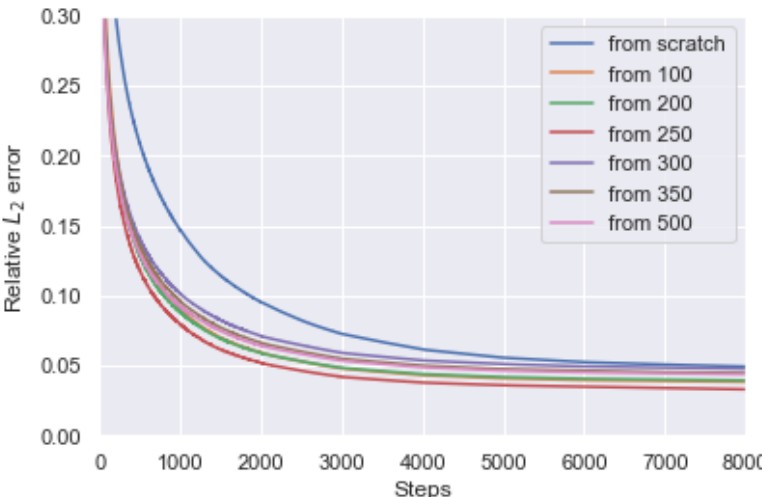

Figure 5: Plot of test relative $L_2$ error versus update step for the Kolmogorov flow with Re500, T=1s. We observe that a ll the operator ansatzs pretrained over PDE instances with different Reynolds number can boost the test-time optimization accuracy and speed compared to training from scratch.

As shown in Figure 6, both LAAF-PINN and SA-PINN converge much faster compared to the original PINN method, but there is still a big gap with PINO. LAAF-PINN adds learnable parameters before the activation function; SA-PINN adds the weight parameter for each collocation point. These techniques help to alleviate the PINNs' optimization problem significantly. However, they didn't alter the optimization landscape effectively in the authors' opinion. On the other hand, by using operator ansatz, PINO optimizes in a function-wise manner where the optimization is fundamentally different.

Note that the contribution of PINO is orthogonal to the above methods. One can apply the adaptive activation functions or self-adaptive loss in the PINO framework too. All these techniques of PINNs can be straightforwardly transferred to PINO. We believe it would be interesting future directions to study how all these methods work with each other in different problems.

## B.2 LID CAVITY FLOW.

We demonstrate an addition example using PINO to solve for lid-cavity flow on $T = [5, 10]$ with $Re = 500$. In this case, we do not have the pre-train phase and directly solve the equation (test-time optimization). We use PINO with the velocity-pressure formulation and resolution $65 \times 65 \times 50$ plus the Fourier numerical gradient. It takes 2 minutes to achieve a relative error of $14.52\%$. Figure 4 shows the ground truth and prediction of the velocity field at $t = 10$ where the PINO accurately predicts the ground truth.

We assume a no-slip boundary where $u(x, t) = (0, 0)$ at left, bottom, and right walls and $u(x, t) = (1, 0)$ on top, similar to Bruneau & Saad (2006). We choose $t \in [5, 10]$, $l = 1$, $Re = 500$. We use the velocity-pressure formulation as in Jin et al. (2021) where the neural operator output the velocity field in $x$, $y$, and the pressure field. We set $width = 32$, $mode = 20$ with learning rate $0.0005$ which decreases by half every 5000 iterations, 5000 iterations in total. We use the Fourier method with Fourier continuation to compute the numerical gradient and minimize the residual error on the velocity, the divergence-free condition, as well as the initial condition and boundary condition. The weight parameters $(\alpha, \beta)$ between different error terms are all chosen as 1. Figure 4 shows the ground truth and prediction of the velocity field at $t = 10$ where the PINO accurately predicts the ground truth.

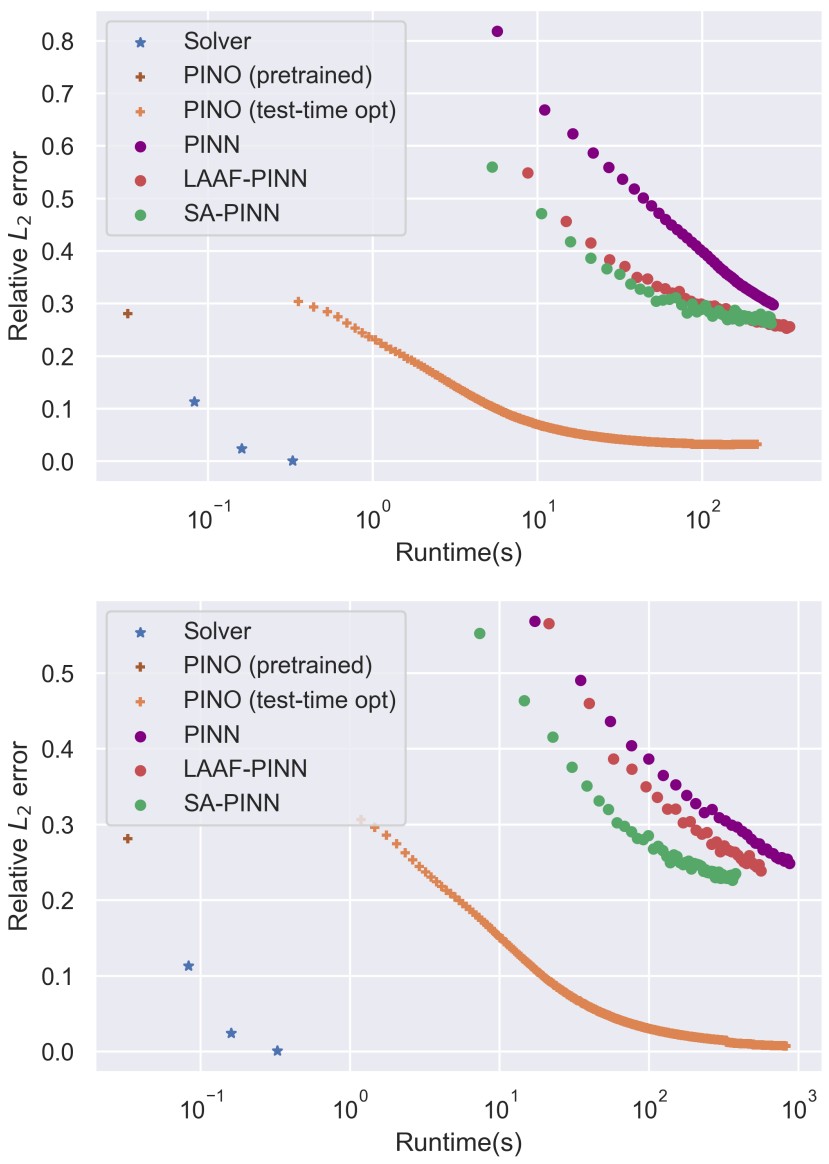

Figure 6: Plot of test relative $L_2$ error versus runtime step for the Kolmogorov flow with Re500, T=0.5s. top: $64 \times 64$, bottom: $128 \times 128$. Averaged over 20 instances.

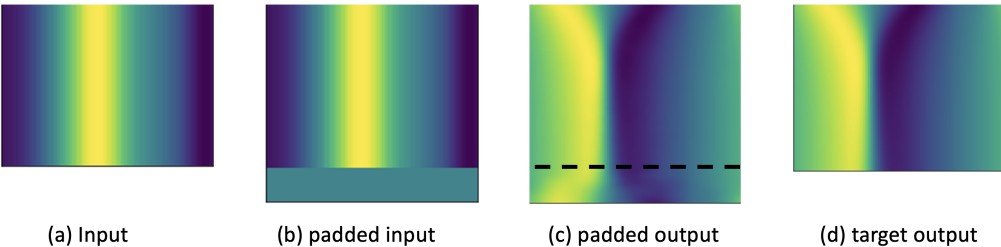

| (a) Input | (b) padded input | (c) padded output | (d) target output |

Figure 7: Fourier Continuation by padding zeros. The x-axis is spatial dimension; the y-axis is the temporal dimension. FNO extends the output smoothly on the padded domain.

## C  FOURIER CONTINUATION

The Fourier neural operator can be applied to arbitrary geometry via Fourier continuations. Given any compact manifold $\mathcal{M}$, we can always embed it into a periodic cube (torus),

$$i : \mathcal{M} \to \mathcal{T}^n$$

where we can do the regular FFT. Conventionally, people would define the embedding $i$ as a continuous extension by fitting polynomials (Bruno et al., 2007). However, in Fourier neural operator, it can be simply done by padding zeros in the input. The loss is computed at the original space during training. The Fourier neural operator will automatically generate a smooth extension to do padded domain in the output, as shown in Figure 7.

This technique is first used in the original Fourier neural operator paper (Li et al., 2020a) to deal with the time dimension in the Navier-Stokes equation. Similarly, Lu et al. (2021b) apply FNO with extension and interpolation on diverse geometries on the Darcy equation. In the work, we use Fourier continuation widely for non-periodic boundary conditions (Darcy, time dimension). We also added an example of lid-cavity to demonstrate that PINO can work with non-periodic boundary conditions.

Furthermore, this Fourier continuation technique helps to take the derivatives of the Fourier neural operator. Since the output of FNO is always on a periodic domain, the numerical Fourier gradient is usually efficient and accurate, except if there is shock (in this case, we will use the exact gradient method).

## D  CONCLUSION AND FUTURE WORKS

In this work, we develop the physics-informed neural operator (PINO) that bridges the gap between physics-informed optimization and data-driven neural operator learning. We introduce the pre-training and test-time optimization schemes for PINO to utilize both the data and physics constraints. In the pre-training phase, PINO learns an operator ansatz over multiple instances of a parametric PDE family. The test-time optimization scheme allows us to take advantage of the learned neural operator ansatz and solve for the solution function on the querying instance faster and more accurately.

There are many exciting future directions. PINO can be used to enhance operator-learning with some constraints, for examples: (1) Control problem: one first learns a solution operator (pre-training) and does test-time optimization in the control phase to make sure it satisfies desired constraints (Hwang et. al.). (2) Multiscale modeling: in multi-scale modeling, one learns a fine-scale solution operator, which is then regulated by a coarse-scale solver. Note that the fine-scale problems do not follow Gaussian distribution as in the training dataset (Liu et. al.). (3) Weather prediction: beyond the observation dataset, one can add physics constraints such as the conservation of mass (Jiang et. al.). (4) Airfoil design: the design problem can be formulated as an inverse problem as shown in Section 4.3. PINO backward formulation is particularly effective for the inverse problem (Thuerey et. al.).

PINO is also designed to work on most of the working scenarios of PINNs: if the optimization is benign, then pre-train is not needed. The major advantage compared to PINN-based methods is the optimization landscape. The operator ansatz, even randomly initialized, still has a much better

optimization landscape compared to the PINN-based model. Most of the techniques and analysis of PINN can be transferred to PINO.

It is interesting to ask how to overcome the hard trade-off of accuracy and complexity, and how the PINO model transfers across different geometries. Furthermore, we can develop a software library of pre-trained models. PINO's excellent extrapolation property allows it to be applied on a broad set of conditions, as shown in the Transfer Reynold's number experiments.

