# OpenReview forum: "Physics-Informed Neural Operator for  Learning Partial Differential Equations"
_ICLR.cc/2022/Conference — ICLR 2022 Submitted_

### Official Review · Reviewer_Tav6 · 2021-10-26

**Correctness:** 2
**Technical Novelty And Significance:** 1
**Empirical Novelty And Significance:** 1
**Recommendation:** 3
**Confidence:** 2

**Main Review:**

Update after author response: The author's responses do not address my concern on the novelty w.r.t. previous papers. Each of the 4 new aspects w.r.t. the previous method is too incremental given the standard knowledge in our communities. Each point is done in other papers without marking them as a contribution of the paper. Also, the comparison should be done with the comparable previous methods as well as better variants of PINNs, as I mentioned in my review. Accordingly, I cannot raise the score to be fair with other papers. I support this direction and I would personally like to recommend acceptance if there is no almost identical idea in the previous paper. However, there is an almost identical idea in the previous paper and I cannot recommend this for acceptance for that reason and additional reasons, including but not limited to the lack of fair comparisons with variants of PINNs.

The proposed method makes sense as a transfer training method for PINNs, with many source tasks and a single target task, although it is limited for the transfer within each specific parametric PDE family. Here, note that this can be viewed as a type of transfer “training” (which is to accelerate training), instead of transfer “learning” (which is designed to accelerate training and generalization).

In this regard, I like the paper and I think that the proposed method has a potential to be used as a transfer training method for PINNs.

However, I have several doubts. First of all, this seems to be too incremental given the following previous work:

[1] Sifan Wang, Hanwen Wang, Paris Perdikaris. 2021. Learning the solution operator of parametric partial differential equations with physics-informed DeepOnets.

The authors should elaborate the difference more. If we replace DeepOnet by FNO in the previous method, does it result in something very similar to PINO?

Moreover, if we know the PDE constraints, I don’t understand the point of learning the solution operator with the PDE constraints in the first phase. If we already know the PDE constraints, we can solve for the solution, without learning the solution operator, for each instance of the PDE family. Thus, we are introducing additional, potentially unnecessary step, in most cases. The authors should rewrite the paper much more carefully to organize the problem setting with what is given and what is not.

From the current presentation, I infer that mathematically, this problem setting is nothing different from that of PINNs for the forward problems, if we don’t have the additional data for the operator learning phase. In this case, from the abstract view point, the proposed method is equivalent to (1) change the architecture from PINNs to FNO for the solution operator, and (2) add additional the pre-training step to train the solution operator for the PDE family, instead of one instance from the PDE family. Thus, the experiments are not convincing, because the performance improvements can be coming from the change of the architecture and the pre-training, which is expected and trivial. The authors mention that PINO is more expressive and thereby achieves better accuracy. However, then, we can also increase the expressive power of PINNs by changing architectures of PINNs. Thus, the authors must compare the proposed method, PINO, against various architectures of PINNs with various expressive powers.

Similarly, to verify the claim of the paper, the paper must include variants of PINNs that have solved many drawbacks of PINNs, such as extended PINNs. At this point, I don’t see any fundamental reason why PINO against variants of PINNs with various architecture choices.

Although the authors mention the limitation of optimization of PINNs, there are various methods proposed to mitigate such limitations, including self-adaptive PINNs (Self-Adaptive Physics-Informed Neural Networks using a Soft Attention Mechanism), adaptive activation functions (Adaptive Activation Functions Accelerate Convergence in Deep and Physics-informed Neural Networks), and sf-PINN (Learning in Sinusoidal Spaces with Physics-Informed Neural Networks). Considering the purpose of the proposed method, the authors must compare PINO against those methods in the paper.

Moreover, for the pre-training, the papers should measure the time complexity of pre-training too to compare it fairly.  My worry is that under this problem setting, we can more efficiently solve each problem instance if we count the extra time of pre-training as well, if I use appropriate architectures and hyper-parameters. Thus, a comprehensive investigation on this is required before publication.

Another question is whether the model pre-trained with one PDE family can generalize well to other different PDE families. This is not conclusive from this paper as well. If we cannot generalize in this way, the applicability of PINO is crucially limited, as we can always solve each instance within a PDE family with variants of PINNs or other PDE solvers in the problem setting of this paper.





**Summary Of The Paper:**

This paper proposes Physics-Informed Neural Operator (PINO) by combining two previous methods, Fourier neural operators (FNO) and physics-informed neural networks (PINNs). Similar to one use case of FNO, PINO learns the solution operator over multiple instances of a parametric PDE family in the first phase. In the second phase, given an instance from the parametric PDE family, PINO fine-tunes the learned solution operator as a pre-trained model to work well for the particular instance. The proposed method is evaluated by using numerical experiments.

**Summary Of The Review:**

This seems to be too incremental given the previous work and the experimental setups are not sufficient to support the claims of the paper. There is no theory provided.

---

> ### Author Response · Authors · 2021-11-20
> **Response to Reviewer Tav6**
>
> Thanks the reviewer for the critical comments. We will address your questions and suggestions:
>
> ### 1. Q: The difference between PINN-DeepONet and PINO
>
> The physics-informed DeepONet and PINO have different goals and problem settings. Previous works such as PINN-DeepONet (Wang et. al.) and Physics-constrained DL modeling (Zhu et. al.) use the PDE constraints in operator learning (corresponding to the pre-training phase in PINO.) On the other hand, **the goal of PINO is to use pre-trained operator ansatz to overcome the optimization challenges in PINN and the generalization issue in operator learning,** none of these was addressed in PINN-DeepONet. PINO is the first work, to our knowledge, that utilizes operators ansatz in equation solving. We propose the **pre-train and optimize** framework, which not just applies PINN-loss to FNO, but fundamentally integrates the two problems settings of operator learning and equation solving. The PINO framework can be extended and applied to other operator models such as DeepONets (if we pre-train the branch net and optimize the trunk net,) which we believe will be an interesting future direction.
>
> Numerically, we do include a comparison between PINN-DeepONet and PINO (just pre-train) on the Burgers equation for learning operators (page 8, first paragraph), where PINN-DeepONet get 1.38% (as reported in the Wang et. al.'s paper) and PINO gets a 0.37% error.
>
> Technically, we propose several methodological advances as well as extensive experiments to understand the optimization and generalization challenges.  Our methodological advances include:
> 1.Instance-wise fine-tuning at test-time to further improve the fidelity of the operator ansatz.
> 2.Novel formulation for inverse problems that result in accurate recovery as well as good speedups.
> 3.Efficient Fourier-space methods for computing derivatives present in the PDE loss.
> 4.Efficient learning through the design of data augmentation and loss functions.
> We now elaborate on each of the above points.
>
> 1. We are the first to formulate PDE learning in two phases: pre-training and fine-tuning. This way we are able to simultaneously overcome the optimization challenges in PINN as well as generalization challenges in operator learning (i.e deeponet or FNO). Also, with this, we can control the accuracy level based on the number of iterations of fine-tuning. Thus, we have a flexible framework where we can tradeoff acuracy with speed for a ML method, like we do with numerical solvers.  In the revised version, we provide these tradeoff plots comparing PINO with other methods including numerical solvers. As shown in Figure 2, PINO has a much better convergence rate compared to PINN.
>
> 2. We are the first to formulate inverse problems as a combination of pre-training and fine-tuning, as well as either forward or backward operator. By directly learning the backward operator, we can obtain the inverse solution directly, without needing to use MCMC with a forward solver. This results in both speedups as well as accurate recovery (in cases like the Darcy flow problem in the paper where the inverse problem is not very ill-posed).
>
> 3. We are the first to develop efficient Fourier-space methods to compute derivatives present in the PDE loss for the Fourier neural operator. In our setting, the input is the full-field (coefficients at all the locations). So computing gradients through autograd is too memory-intensive and infeasible. By developing Fourier-space methods, we make PINO practical for solving challenging PDE families.
>
> 4. We also study the role of augmentation techniques and loss design. Since obtaining ground-truth solutions from numerical solvers is expensive, we augment instances with only PDE loss during pre-training, which can be done without any ground-truth solutions. This improves the generalization capabilities of the operator ansatz. For fine-tuning at test-time on a given instance, we   propose a loss that combines PDE loss of the instance with operator-learning loss from pre-training to avoid overfitting and catastrophic forgetting. While similar methods have been developed for computer vision problems, this is the first time they are studied for PDE problems.

---

> > ### Author Response · Authors · 2021-11-20
> > **Response to Reviewer Tav6 (part 2)**
> >
> > ### 2. Q: why pre-train if the PDE constraints are known?
> >
> > #### Pre-training makes previously unsolvable problems solvable
> >
> > PINNs face several optimization issues: (1) the challenging optimization landscape from soft physics or PDE constraints, (2) the difficulty to propagate information from the initial or boundary conditions to unseen parts of the interior or to future times, and (3) the sensitivity to hyper-parameters selection. For example, in the long-temporal transient flow example (Section 4.1), none of the methods can achieve a reasonable accuracy since it is extremely challenging to propagate the information from the initial condition to future time steps over such a long interval T = [0, 50] just using the soft physics constraint. However, when the pre-training data is available for PINO, we can use the learned neural operator ansatz and the operator loss. **The pre-trained operator loss is a direct constraint that makes the optimization much easier.** In this case, the model does not need to propagate information from the initial to a future time, and the method becomes much robust to the choices of hyperparameters. In the end, pretrain operator ansatz gets a 1.84% error rate, while other method fails to converge.
> >
> > #### Pretraining is optional for PINO to outperform PINNs.
> > The major advantage compared to PINN-based methods is the optimization landscape. **The operator ansatz, even randomly initialized, still has a much better optimization landscape compared to the PINN-based model.** As shown in experiments of the Kolmogorov flow (Section 4), the PINO without any pre-training still gets much better performance compared to PINNs baselines.
> >
> > The operator ansatz has a better optimization landscape (even without pretraining) because it does function-wise optimization while PINN optimizes in a pointwise manner. The neural operator parameterizes the solution function as an aggregation of basis functions, and hence, the optimization is in the function space. This is easier than just optimizing a single function as in PINN.
> >
> > ### 3. Q: significance of the operator ansatz.
> > We add a comparison experiment against the Locally adaptive activation functions for PINN [(LAAF-PINN)](https://github.com/antelk/locally-adaptive-activation-functions) and Self-Adaptive PINN [(SA-PINN)](https://github.com/levimcclenny/SA-PINNs), as suggested by the reviewer. For the Kolmogorov flow problem, with Re=500, T=[0, 0.5], averaged over 20 testing instances. (Please see Appendix B for the details.)
> >
> > ![](https://i.imgur.com/Jgql3xC.png)
> >
> > | Computation budget (s) | 0.03s | 1s | 10s | 50s |100s |
> > | -------- | -------- | -------- |-------- | -------- |-------- |
> > |PINO| 28.12% | 23.12%| 7.00%| 3.69%    |3.22%
> > |PINN| -| -|66.89% | 47.25% | 39.68%|
> > |LAAF-PINN|- |- | 54.87%| 33.34%| 29.38%|
> > |SA-PINN|- | - | 47.13%| 30.46%| 29.65%|
> >
> > **As shown in the figure above, both LAAF-PINN and SA-PINN converge much faster compared to the original PINN method, but there is still a big gap with PINO.** After converged, none of these PINN variations reach the **20%** relative L2 error rate, while PINO achieves an **1%** error rate. The LAAF-PINN adds learnable parameters before the activation function; SA-PINN adds the weight parameter for each collocation point. These techniques help to alleviate the PINNs' optimization problem significantly. However, they didn't alter the optimization landscape effectively in the authors' opinion. On the other hand, by using operator ansatz, PINO optimizes in a function-wise manner where the optimization is fundamentally different.
> >
> > Note that the contribution of PINO is orthogonal to the above methods. One can apply the adaptive activation functions or self-adaptive loss in the PINO framework too. **All these techniques of PINNs can be straightforwardly transferred to PINO.** We believe it would be interesting future directions to study how all these methods work with each other in different problems.

---

> > > ### Author Response · Authors · 2021-11-20
> > > **Response to Reviewer Tav6 (part 3)**
> > >
> > > ### 4. Q: Complexity of pre-training:
> > > In most of our examples, including the Burgers, Darcy, and long-temporal Navier-Stokes equation, the pre-training only takes about **1-2 hours** with a single Nvidia V100 GPU. The most expensive training process (Kolmogorov Flow 400+400k) takes no more than **10-20 hours** on a single GPU. Considering the PINN-based method on average takes about 10 minutes for each instance, it will be worthy to pre-training if one wants to solve for more than about 100 instances.
> > >
> > > We want to emphasize that the pre-training phase can be done offline. Once trained, such pretrained operators can be generalized to a wide class of problems as discussed above. Similar to numerical software, the pre-trained model is distributed publically for everyone to use. Therefore **we would like to consider the training time as part of solver-design.** Compared to the days and months researchers build up numerical solvers, 20 hours of training time is not too bad.
> > >
> > > ### 5. Q: generalization to new distribution:
> > > Generalization is one of the major motivations of our work. Standard operator learning as a supervised learning task usually assumes a data distribution, which is not always accessible for the real world. **By introducing test-time optimization, we overcome the assumption of distribution.** As shown in the Reynolds-transfer experiment (Figure [5] and Table [5]), the operator trained from different Reynolds numbers (corresponding to different distributions) can be easier transferred to each other with a small amount of test-time optimization. This transferability relieves the distribution assumption and makes the PINO framework much more flexible.
> > >
> > >
> > > We hope that the reviewer found our responses and additional comparison studies insightful. If the reviewer has any other concerns or wants any clarifications, please do not hesitate to let us know! If the Reviewer is satisfied with our method’s contributions and analyses, we hope that the reviewer could increase our review score as appropriate.

---

### Official Review · Reviewer_oyPM · 2021-10-29

**Correctness:** 4
**Technical Novelty And Significance:** 3
**Empirical Novelty And Significance:** 3
**Recommendation:** 5
**Confidence:** 2

**Main Review:**

Given the claims of novelty is correct (that this paper is the first one proposing the combination of PINN and FNO) (I am not that familiar with the relevant latest literature there), I think this is a great paper with both technical contribution and strong emperical result.

Strength: The idea to combine these two mainstream pde solver is, although simple, very effective as shown from the emperical results. The only technical difficulty of combining the two algorithm (test time PINN-based optimization for FNO) that I see is the derivative calculation for neural operator space, where the authors used exact gradient calculation coded by chainrule. The loss landscape

Weakness/Questions: Some of the questions and rooms for potential improvement:
1. Fig 2c the colors are very hard to recognize on the legend.
2. Did the PINN benchmark in this work include the adaptive activation function? ("Adaptive activation functions accelerate convergence in deep and physics-informed neural networks" Jagtap & Karniadakis.)
3. According to Table 2 for the Kolmogorov flow, the PINO have 0 solution error improvement with increase in both number of data samples and PDE instances. Although the test time optimization convergence is faster, didn't it take much longer time to pre-train in these two cases?
4. The clarity for the section 4.3 inverse problem needs to be improved (maybe due to page limit?). For the forward model method, how many initializations of "a" were tried? What was the R(a) regularization term's underlying assumption there?

**Summary Of The Paper:**

The authors presented a novel algorithm to solve PDEs by combining two mainstream PDE solving deep neural network genre, the PINN and the FNO method into PINO. Which is basically doing these two steps sequentially by first learning the neural operator and then do a optimization step using PINN method.

**Summary Of The Review:**

I think the technical and algorithmic contribution is solid ,the emperical results are good, and (if the novelty is good) I think this is a publishable paper.

---

> ### Author Response · Authors · 2021-11-20
> **Response to Reviewer oyPM**
>
>
> We want to thank the reviewer for the thoughtful feedback. We will address the questions and suggestions in the following:
> ### 1. Q: the color of the figure
> Thank the reviewer for the suggestion. We have updated figure [2], where we plot the accuracy-complexity plots for the long-temporal NS equation and Chaotic KF equation in log-log scale.
>
> ### 2. Q: additional baselines
> We add a comparison experiment against the Locally adaptive activation functions for PINN [(LAAF-PINN)](https://github.com/antelk/locally-adaptive-activation-functions) and Self-Adaptive PINN [(SA-PINN)](https://github.com/levimcclenny/SA-PINNs), as suggested by the reviewers. For the Kolmogorov flow problem, with Re=500, T=[0, 0.5]
>
> (Please see Appendix B and Figure 6 in the revised paper.)
>
> ![Comparison on resolution 64x64x65](https://i.imgur.com/Jgql3xC.png)
>
> ![Comparison on resolution 128x128x65](https://i.imgur.com/rpYtbTL.png)
>
> | Computation budget (s) | 0.03s | 1s | 10s | 50s |100s |
> | -------- | -------- | -------- |-------- | -------- |-------- |
> |PINO| 28.12% | 23.12%| 7.00%| 3.69%    |3.22%
> |PINN| -| -|66.89% | 47.25% | 39.68%|
> |LAAF-PINN|- |- | 54.87%| 33.34%| 29.38%|
> |SA-PINN|- | - | 47.13%| 30.46%| 29.65%|
>
> **As shown in the figure above, both LAAF-PINN and SA-PINN converge much faster compared to the original PINN method, but there is still a big gap with PINO.** LAAF-PINN adds learnable parameters before the activation function; SA-PINN adds the weight parameter for each collocation point. These techniques help to alleviate the PINNs' optimization problem significantly. However, they didn't alter the optimization landscape effectively in the authors' opinion. On the other hand, by using operator ansatz, PINO optimizes in a function-wise manner where the optimization is fundamentally different.
>
> Note that the contribution of PINO is orthogonal to the above methods. One can apply the adaptive activation functions or self-adaptive loss in the PINO framework too. **All these techniques of PINNs can be straightforwardly transferred to PINO.** We believe it would be interesting future directions to study how all these methods work with each other in different problems.
>
> ### 3. Q: Pre-training
> As shown in Table [3] in the revised version. PINO, both with/without pretrain, can reach the <1% error. It suggest that
> **if the optimization is benign, then pre-training can be optional.** The major advantage compared to PINN-based methods is the optimization landscape. The operator ansatz, even randomly initialized, still has a much better optimization landscape compared to the PINN-based model. Further discussions of pretrain can be found in the common response above.
>
> In this experiment, the pre-training phase takes no more than **10-20 hours** on a single GPU. Considering it speeds up about 2 minutes for each instance, it will be worthy to pre-training if one wants to solve for more than 600 instances. We also want to emphasize that the pre-training phase can be done offline. Once trained, such pretrained operators can be generalized to a wide class of problems as discussed above. Similar to numerical software, the pre-trained model is distributed publically for everyone to use. Therefore **we would like to consider the training time as part of solver-design.** Compared to the days and months researchers build up numerical solvers, 20 hours of training time is not too bad.
>
>
> ### 4. Q: Write-up of Section 4.3 Inverse problem
> A: Thanks for understanding. We have revised the section of the inverse problem in the updated version.

---

> > ### Comment · Reviewer_oyPM · 2021-11-25
> > **Thanks for the rebuttal!**
> >
> > Thank you authors for your detailed rebuttal, those addressed nearly all of my concerns.
> >
> > After checking the PINN-DeepONet (Wang et. al.) paper, I agree with reviewer Tav6 that the overall model structure is just too incremental, which is very different from what I've read from the manuscript previously.
> >
> > However, I also agree with the authors that the tweak from "using the PINN as regularizer during training" versus "using the PINN as optimizer for fine-tuning" during inference, although architecturally trivial, have shown its applicability and promise in solving these problems in a unique way.
> >
> > Given the above, I have updated my scores to 5, marginally below acceptance. I think one possible way of improvement for this paper is to emphasis its difference and advantage of separating the PINN with the operator learning stage by more ablation study or visualizations directly to support the argument of doing so helps with the optimization challenges.

---

> > > ### Author Response · Authors · 2021-11-26
> > > **Thank reviewer oyPM for the response!**
> > >
> > > We thank the reviewer for the prompt feedback. We are glad to see that most of the concerns have been addressed. We especially appreciate the reviewer agrees that using operator ansatz “has shown its applicability and promise in solving these problems in a unique way.”
> > >
> > > We would like to take the opportunity to elaborate on our **technical and architectural contributions**. Using operator ansatz in PINO brings a significant optimization advantage over the standard neural network ansatz in PINNs. Because the operator takes the full field function as input and output, it does function-wise optimization while both PINN and PINN-DeepONet (trunk net) optimize in a pointwise manner. This also makes it necessary for us to develop novel efficient gradient computation methods. Our results show that this makes optimization much easier compared to optimizing a single function as in PINN, which is shown in the figure:
> > >
> > > ![128x128](https://i.imgur.com/rpYtbTL.png)
> > >
> > > We now provide more details about the efficient **function-wise gradient computation**. The function-wise optimization in PINO brings in a non-trivial challenge to compute the gradient in a function-wise manner. Standard PINN and PINN-DeepONet (if pretrain the branch net and optimize the trunk net) are pointwise functions u: x->u(x), which allows straightforward adaption of the auto differentiation in TensorFlow and Pytorch to access du/dx. However, pointwise optimization brings computation challenges. On the other hand, the operator ansatz used in PINO takes function-wise input G: a->u. One cannot directly apply the PINN (autograd method) to compute du/dx.
> > >
> > > To overcome these technical challenges, we developed effective methods to compute the gradient in the function-wise manner, as discussed in section 3.3. We develop efficient Fourier-space methods to compute derivatives present in the PDE loss for the Fourier neural operator. Since FNO does the computation on the Fourier space, we can compute the exact gradient in the Fourier space. In the implementation, we implemented the gradient on the Fourier space, the off-grid evaluation functions, the batched chain rule computation. These modifications do not impact the FNO in the inference, but affect the gradient and loss function. The function-wise gradient is key to function-wise optimization.
> > >
> > > Further, we formulate the **Fourier continuation** technique. Standard FFT can be only applied on periodic, square domains. To compute the derivatives with the Fourier method on any domain, we embed it into a periodic cube by padding zeros. The Fourier neural operator will automatically generate a smooth extension to the padded space in its output, as shown in the figure below. In this case, the Fourier gradient is usually efficient and accurate.
> > >
> > > ![Fourier continuation](https://i.imgur.com/hwRMslr.png)
> > >
> > > As suggested by the reviewer, we would like to add more visualizations directly to support our argument that using pre-trained operator ansatz helps with the optimization challenges and generalization:
> > >
> > > (1) Optimization challenges: the figure shows the prediction of PINN and PINO on Kolmogorov flow with Re=500. As shown in the figure, PINO (even without pre-training) has significantly better accuracy and details due to the operator ansatz.
> > >
> > > ![PINO vs PINN baselines](https://i.imgur.com/H8DwFQg.jpg)
> > >
> > >
> > > (2) Reynolds number: In in Reynolds-transfer experiment, we pre-train the operator ansatz on Re=100 data and fine-tune on the Re=500 problem. Despite the Re=100 and Re=500 flows having quite different structures, PINO is able to generalize and extrapolate from one to another
> > >
> > > ![Transfer Reynolds number](https://i.imgur.com/UVqJYbR.png)

---

### Official Review · Reviewer_WVp6 · 2021-11-01

**Correctness:** 3
**Technical Novelty And Significance:** 2
**Empirical Novelty And Significance:** 2
**Recommendation:** 6
**Confidence:** 3

**Main Review:**

The paper is well-written and highlights the current challenges in PDE learning before introducing the approach. The following paragraphs detail the strengths and weaknesses of the paper.

Strengths
---
- The paper is well-written and structured. The authors gives a good introduction of the field with its challenges and describe their contributions clearly. The next sections describe the PINN and FNO approaches before introducing the authors novelty method called PINO. The authors conclude by presenting a series of experiments.
- The authors introduce a method that combines the strengths of two popular approaches in the field of physics-informed machine learning: a stable and fast method for learning solution operators based on a large number of input-output training pairs (FNO) and a deep learning technique for solving PDEs using PDE constraints. The main contribution consists of applying the physics (PDE) constraints onto a trained neural operator. At the heart of this technique lies an efficient procedure for computing derivatives of Fourier neural operator described in section 3.3 of the paper.
- The authors show experiments on challenging (nonlinear, two-dimensional, chaotic, time-dependent) examples and achieve better results than standard deep learning methods on the three setups that they consider: learning solution operators, solving PDEs, solving inverse problems.

Weaknesses
---
- My main concern about the method described is that it requires the knowledge of the PDE expression and therefore the title is a bit misleading since the PDE is already known and hard-coded as a constraint in the optimization. Therefore, it also combines the two weaknesses of FNO and PINNS: the need for training input-output data and the knowledge of the PDE, this should be explicitly written in the conclusions of the paper as a limitation.
- I have some concerns about the choice of the training and testing data and optimization parameters. The training data used by the authors seem to be optimized for the problem considered, which questions the stability of the method. Is it able to tackle a new application without a lot of fine-tuning ? As an example, the Gaussian kernel used to sample the initial conditions vary across all the examples considered following a logic that I don't understand (see the experiments details described in the Appendix). This is a large limitation of the proposed method since it's heavily affects its flexibility for new applications. Similarly, the authors give little details about the training test used in the different applications. I am then wondering whether the good performances shown by the authors is due to the fact that the testing and training sets for the forcing terms (or initial conditions) are drawn from the same random distribution. In this case, the method could only be learning the solution operator on a specific vector space of functions and not in general (i.e. transfer learning would not work on a completely different distribution).
- I don't think that the authors report an example with noisy data for learning and solving PDEs, which is essential to check the robustness of the method.
- To my understanding, the Fourier neural operator method has severe limitations on the type of domains and boundary conditions, unlike PINNs (Raissi et al., Science, 2020), as it requires simple geometries and periodic boundary conditions to apply the Fourier transform layers. These limitations carries over in the present paper and the method can only be applied on simple domains with homogeneous/periodic boundary conditions. This questions the practical utility of the method for e.g. solving PDEs with respect to traditional numerical solvers such as FEM/spectral methods/finite differences. Can the authors comment on that, point out the limitations in the conclusions of the papers and discuss potential ways to overcome them ?
- It would be great if the authors could add a paragraph to give a hint on potential practical concrete applications of the present methods, where one would have access and take advantage of both training data (sampled from a carefully selected Gaussian distribution) and the expression of the PDE. Right now, I can hardly see any concrete applications beyond fluid dynamics problems.

Minor comments
---
Can the authors explain what they mean by "additional PDE instances" in Table 1 ? I couldn't find it in the Appendix and this should be described explicitly in the main text.

### Post-rebuttal update
I updated my score following the rebuttal.

**Summary Of The Paper:**

The authors introduce physics-informed neural operator for learning partial differential equations (PDEs) from data with deep learning. This approach combines two recent and popular methods in the field of machine learning and PDEs, the physics-informed neural network (PINNs) for solving PDEs and Fourier neural operator (FNO) for learning solution operators. The paper takes advantages of both approaches to overcome convergence issues arising in PINNs by incorporating PDE constraints into FNO. Then, the authors apply their method to a number of challenging test cases and obtain better results than PINNs on PDE solving problems and FNO for learning solution operators.

**Summary Of The Review:**

I find that the paper is well-written and introduces a method combining two known techniques for learning and solving PDEs from data. The authors present numerical results that outperform PDE benchmark used in the literature, taking advantage of the PDE constraint from the PINN approach or the neural operator learning from the FNO technique. Yet, there are some weaknesses in the study that I discuss in the main review. However, most weaknesses are inherent from the current challenges faced by the field of PDE discovery (such as choice of training data, practical applications, limitations on the domain geometry, boundary conditions).

Therefore, I believe that the paper could motivate further explorations and research between deep learning PDE solvers and PDE learning methods and recommend acceptance of the paper to ICLR provided that the authors address the points raised in the review.

---

> ### Author Response · Authors · 2021-11-20
> **Response to Reviewer WVp6 (part 1)**
>
> We are encouraged by the highly positive comments from the reviewer. These constructive suggestions help us improve our work. Regarding the questions:
>
> ### 1. Q: **Requirement on the explicit form of equation**:
> We agree that the stronger performance of PINO is based on the extra knowledge of the equation compared to FNO. We would like to think our work completes the big picture of operator learning: it offers a solution for any scenarios (given data and/or equation constraints, operator learning, and/or equation solving). We would like to emphasize that data and equations can be optional.  Without any data or pretraining, PINO still strongly outperforms PINN baselines. In practice, the data can be partially observed and the physics constraints may not be complete. One just uses whatever is available.
> ### 2. Q: **Generalization to new distribution**:
> Generalization is one of the major motivations of our work. Standard operator learning as a supervised learning task usually assumes a data distribution, which is not always accessible for the real world. **By introducing test-time optimization, we overcome the assumption of distribution.** As shown in the Reynolds-transfer experiment (Figure [5] and Table [5]), the operator trained from different Reynolds numbers (corresponding to different distributions) can be easier transferred to each other with a small amount of test-time optimization. This transferability relieves the distribution assumption and makes the PINO framework much more flexible.
> ### 3. Q: Robustness:
> A: Regarding robustness. We add extra experiments to study the robustness of neural operators on the Burgers, Darcy, and Navier-Stokes equation in the full data setting (similar to FNO). We consider two scenarios (1) train with clean data and (2) train with noisy data. The noise is added on the input via
>
> 	x = x + e, where e ~ a * max(x) * N(0,1)
>
>
> |         | Training error | Testing (clean) | Testing (10% noise) |
> |---------|----------------|-----------------|---------------------|
> | Darcy |        0.6%        |      1.1%           |       1.2%              |
> | Burgers   |    0.2%            |      0.2%           |       3.8%              |
> | NS      |      2.4%          |         2.4%        |         3.9%            |
> | Darcy  (train with noise)|       0.7%         |       1.2%          |      1.2%               |
> | Burgers  (train with noise) |     2.0%           |      2.0%           |        2.0%             |
> | NS     (train with noise) |       2.6%         |        2.6%         |          2.5%           |
>
> In general, we observe the neural operator is robust, as long as it's not severely overfitting to the training set. The performance of the neural operator degrades when adding >=10% noise. The Darcy equation is most robust to noise since it has a piecewise constant structure. However, the performance of the Burgers and Navier-Stokes equation degrades when training on the clean dataset. Training with noise will make the model more robust. In this case, there is no domain shift and the operator becomes robust to noise, but it may come with the cost of the loss of precision.
> ### 4. Q: non-periodic, non-uniform geometry:
> The Fourier neural operator can be applied to arbitrary geometry via **Fourier continuation**. Given any compact manifold $\mathcal{M}$, we can always embed it into a periodic cube (torus), $i: \mathcal{M} \to \mathcal{T}^n$, where we can do the regular FFT. Conventionally, people would define the embedding $i$ as a continuous extension by fitting polynomials. However, in Fourier neural operator, it can be simply done by padding zeros in the input. During training, the loss is only computed at the original domain. The Fourier neural operator will automatically generate a smooth extension to the padded space in its output.
>
> ![](https://i.imgur.com/hwRMslr.png)
> Fourier continuations on 1d Burgers equation.
>
> This technique is first used in the original Fourier neural operator paper (Li et. al.) to deal with the time dimension in the Navier-Stokes equation. Similarly, Ming et. al. applied it on [Cylinder flow](https://github.com/foldfelis/NeuralOperators.jl). In the PINO work, we use Fourier continuation widely for non-periodic boundary conditions (Darcy, time dimension). We also added an example of lid-cavity to demonstrate that PINO can work with non-periodic boundary conditions.
>
> Furthermore, this Fourier continuation technique helps to take the derivatives of the Fourier neural operator. Since the output of FNO is always on a periodic domain, the numerical Fourier gradient is usually efficient and accurate, except if there is shock (in this case, we will use the exact gradient method). Please see Appendix-C for details.

---

> > ### Author Response · Authors · 2021-11-20
> > **Response to Reviewer WVp6 (part 2)**
> >
> > ### 5. Q: concrete applications of PINO
> >
> > It seems to us that the reviewer's worry is based on the concerns of generalization and geometry. As we have discussed above, PINO has a strong capability of generalization and extrapolation, and it can be applied to arbitrary geometries. We hope the reviewer's concern is already resolved. We are happy to list a few concrete examples:
> >
> > **Operator-learning with some constraints:**
> > - Control problem: one first learns a solution operator (pre-training) and does test-time optimization in the control phase to make sure it satisfies desired constraints ([Hwang et. al.](https://arxiv.org/abs/2111.04941)).
> > - Multiscale modeling: in multi-scale modeling, one learns a fine-scale solution operator, which is then regulated by a coarse-scale solver. Note that the fine-scale problems do not follow Gaussian distribution as in the training dataset ([Liu et. al.](https://arxiv.org/abs/2102.07256)).
> > - Weather prediction: beyond the observation dataset, one can add physics constraints such as the conservation of mass ([Jiang et. al.](https://arxiv.org/abs/2110.07100)).
> > - Airfoil design: the design problem can be formulated as an inverse problem as shown in Section 4.3. PINO backward formulation is particularly effective for the inverse problem ([Thuerey et. al.](https://arxiv.org/abs/1810.08217)).
> >
> > We also want to emphasize that PINO is designed to work on **most of the working scenarios of PINNs:** if the optimization is benign, then pre-train is not needed. The major advantage compared to PINN-based methods is the optimization landscape. The operator ansatz, even randomly initialized, still has a much better optimization landscape compared to the PINN-based model.

---

> > > ### Comment · Reviewer_WVp6 · 2021-11-28
> > > **Response to the rebuttal**
> > >
> > > Thanks to the authors for the rebuttal, my concerns were mostly addressed except my second point as there is still no justification in the revised manuscript for the choice of initial conditions, whose distribution vary depending on the application considered.
> > >
> > > However, I have read the other reviews and responses and am concerned by the novelty point raised by Reviewer Tav6 as I wasn't aware of the Wang et al. paper. The current version of the manuscript (abstract and our contributions section) does not clarify the novelty with respect to this specific paper. After reading the Wang et al. paper, I realized that the idea of incorporating the PDE knowledge into the operator learning is not new and that the author's main novelty is the pre-training phase to improve the PINN solver. However, this is sort of equivalent to the problem solved by Wang et al.. On the other hand, the authors provide convincing numerical experiments on challenging datasets (operator learning, transfer learning, solving Navier-Stokes equations).
> > >
> > > For the reasons listed above I decreased my score to marginally above the acceptance threshold. One way of improving the paper is to clearly highlight the novelty with respect to the papers combining PINN and Operator Learning (from the current version of the paper, it seems that this is the main novelty), and perform a comparison with the Wang et al. paper to show that the proposed methodological advances are significant. Comparing to PINN-like technique is not a really fair comparison since the proposed technique incorporates additional data: input-output function pairs (a_j,u_j), which assumes the existence of an accurate PDE solver.

---

> > > > ### Author Response · Authors · 2021-11-28
> > > > **Further response to reviewer WVp6**
> > > >
> > > >
> > > > Thank reviewer WVp6 for the response. We especially appreciate the comments on "convincing numerical experiments on challenging datasets". We would like to take this opportunity to emphasize our novelty and technical innovations.
> > > >
> > > >
> > > > ### Comparison with PINN-DeepONet
> > > > There are indeed previous works like PINN-DeepONet, as well as Physics-constrained DL modeling (Zhu et. al.) that use the PDE constraints in operator learning. Compared to them, our two major contributions are to (1) propose the two-phase learning with pre-training and instance-wise optimization framework, and (2) efficient function-wise optimization while the previous works are only point-wise. We now elaborate on these aspects.
> > > > 1. We propose the **pre-training and instance-wise optimization** framework that uses the learned operator ansatz with instance-wise fine-tuning to overcome the optimization challenges in PINN and the generalization challenges in operator learning. In contrast to our work, PINN-DeepONet has only the pre-training phase and suffers from generalization challenges. On the Kolmogorov flow, test-time optimization in our framework  reduces the error from **24.2% to 0.9%** (as shown in Table 2), which is a significant improvement compared to standard operator-learning as in PINN-DeepONet.
> > > > 2. On the architecture level, our framework PINO does function-wise optimization while PINN-DeepONet (trunk net) only optimizes in a pointwise manner (also true for PINN). The function-wise optimization brings significant architecture advantages, which is discussed in the paragraph below. For numerical example, on Burgers equation and standard operator learning scenario, PINN-DeepONet achieves **1.38%** relative l2 error while our framework PINO obtains **0.37%**  error (as shown in Table 1), which is  a significant improvement.
> > > >
> > > > We are happy to add these explanations to our paper to emphasize the differences with previous works.
> > > >
> > > >
> > > > ### Function-wise optimization
> > > > Even though DeepONet-PINN uses the PDE loss, it does only pointwise optimization in our setup of instance-wise finetuning, since we pretrain the branch net and optimize the trunk net for each instance. This can use autograd in a straightforward manner. In contrast, in our framework PINO, the operator takes the full field function as both input and output, it needs function-wise optimization and makes it necessary for us to develop novel efficient gradient computation methods.
> > > >
> > > > The contribution to the function-wise optimization is independent of pre-training. We would like to clarify that our framework PINO, **without any pretraining**, still strongly outperforms variations of PINN and DeepONet-PINN methods.
> > > >
> > > >
> > > > ### Comparison with PINNs
> > > > We would like to emphasize that PINO, **without any pretraining**, still strongly outperforms variations of PINN baselines as shown in the figure below. Because of the function-wise optimization and architecture advantages. Since none of the methods use data or additional training processes, would the reviewer agree this is a fair comparison between PINNs and PINO?
> > > >
> > > > ![128x128](https://i.imgur.com/rpYtbTL.png)
> > > >
> > > >
> > > >
> > > > ### Generalization of the choice of initial conditions
> > > > In the end, we would like to address the reviewer's concern that we need to assume the initial conditions are drawn from a Gaussian random field. This assumption holds in general for operator learning but **it is overcome in PINO**.
> > > > 1. The pretraining process is optional. As shown in the lid-cavity example with Reynolds number 500, PINO accurately solves for the solution without a dataset. The input function is the boundary u(0,1) = 1, is a constant function, not sampled from Gaussian distribution.
> > > > 2. The learned operator ansatz can be easily generalized to different distributions, even if pretrained on the Gaussian random field. This is shown in the Reynolds-transfer experiment.
> > > >
> > > > Therefore, we don't think there is any limitation on the choice of initial conditions. If the reviewer has further concerns please don't hesitate to let us know!

---

### Official Review · Reviewer_NBhE · 2021-11-02

**Correctness:** 3
**Technical Novelty And Significance:** 3
**Empirical Novelty And Significance:** 3
**Recommendation:** 5
**Confidence:** 3

**Main Review:**

 I have read the other reviews and responses from the authors. Two highly-related papers should be added. So the claims and experiments need major revisions. I lower my score to 5.
===========================================

The paper is well-written and organized. The experiments are convincing. I have the following two main concerns, which prevents me from improving the score.
1. For a PDE solver, the data is specialized for the PDE. So the whole solving process should include the pre-training stage.
2. For vision tasks, using the pre-trained models to improve the accuracy and efficiency of the downstream tasks is a promising way. These vision tasks share the features from images. However, for a given PDE solving problem, when one can use the pre-trained model? Do we need to train a lot of pre-trained models?

**Summary Of The Paper:**

The paper proposes the physics-informed neural operator (PINO). It combines the operating-learning and function-optimization
frameworks, which improves convergence rates and accuracy over traditional  methods. Experiments test the advantage of PINO.

**Summary Of The Review:**

The paper is clear and the proposed method is reasonable. It has the weaknesses on the use of pre-trained models for PDE solving. However, I tend to accept the paper for its effectiveness.

---

> ### Author Response · Authors · 2021-11-20
> **Response to Reviewer NBhE**
>
>
> We thank the reviewer for the comments. We are glad that the reviewer agrees that the paper is well-organized and the experiments are convincing. We want to emphasize that our major contribution is to use operator ansatz to overcome the optimization challenge in PINN. Pre-training is optional to outperform PINNs.
>
> The reviewer has two concerns on **pre-training**:
>
> ### 1. Q: **How expensive is the pre-training?**
>
> In most of our examples, including the Burgers, Darcy, and long-temporal Navier-Stokes equation, the pre-training only takes about **1-2 hours** with a single Nvidia V100 GPU. The most expensive training process (Kolmogorov Flow 400+400k) takes no more than **10-20 hours** on a single GPU. Given that the PINN-based methods take about 10 minutes on average for solving each instance, it is worth pre-training if one wants to solve for O(10^2) instances.
>
>
> We want to emphasize that the pre-training phase can be done offline. Once trained, such pretrained operators can be generalized to a wide class of problems as discussed above. Similar to numerical software, the pre-trained model is distributed publically for everyone to use. Therefore **we would like to consider the training time as part of designing the numerical solver.** Compared to the months and sometimes years researchers spend crafting numerical solvers, 20 hours of training time is, in fact, an incredible speed-up.
>
> ###  2. Q: **When to use pre-training?**
> In short, we do pre-training in one of the following cases:
> (i) The problem is too hard to solve without data.
>  (ii) Data/computation is available.
>  (iii) Applications require solving for multiple instances.
>
> #### The problem is too hard to solve without data.
> PINNs face several optimization issues: (1) the challenging optimization landscape from soft physics or PDE constraints, (2) the difficulty to propagate information from the initial or boundary conditions to unseen parts of the interior or to future times, and (3) the sensitivity to hyper-parameters selection. For example, in the long-temporal transient flow example (Section 4.1), none of the methods can achieve a reasonable accuracy since it is extremely challenging to propagate the information from the initial condition to future time steps over such a long interval T = [0, 50] just using the soft physics constraint. However, when the pre-training data is available for PINO, we can use the learned neural operator ansatz and the operator loss. **The pre-trained operator loss is a direct constraint that makes the optimization much easier.** In this case, the model does not need to propagate information from the initial to a future time, and the method becomes much more robust to the choices of hyperparameters. In the end, pretrained operator ansatz gets a 1.84% error rate, while other methods completely fail to converge.
>
> #### Pretraining is optional in PINN's setting
>
> If the problem is simple and the optimization is benign, then pre-train is not needed. The major advantage compared to PINN-based methods is the optimization landscape. **The operator ansatz, even randomly initialized, still has a much better optimization landscape compared to the PINN-based model.** As shown in experiments of the Kolmogorov flow and Cavity flow (Appendix B), the PINO without any pre-training still gets much better performance compared to PINNs baselines.
>
> The operator ansatz has a better optimization landscape (even without pretraining) because it does function-wise optimization while PINN optimizes in a pointwise manner. The neural operator parameterizes the solution function as an aggregation of basis functions, and hence, the optimization is in the function space. This is easier than just optimizing a single function as in PINN.

---

> ### Author Response · Authors · 2021-11-29
> **Further response to reviewer NBhE**
>
> Dear reviewer NBhE, we have seen you update your comment. Could you clarify which two papers should be added? If you mean PINN-DeepONet (Wang et. al.) and Physics-constrained DL modeling (Zhu et. al.), please see our response in the first paragraph. If you mean LAAF-PINN and SA-PINN, please see the second paragraph.
>
> ### 1. Comparison with PINN-DeepONet and physics-constrained DL modeling
> There are indeed previous works like PINN-DeepONet, as well as Physics-constrained DL modeling (Zhu et. al.) that use the PDE constraints in operator learning. Compared to them, our two major contributions are to (1) propose the two-phase learning with pre-training and instance-wise optimization framework, and (2) efficient function-wise optimization while the previous works are only point-wise. We now elaborate on these aspects.
> 1. We propose the **pre-training and instance-wise optimization** framework that uses the learned operator ansatz with instance-wise fine-tuning to overcome the optimization challenges in PINN and the generalization challenges in operator learning. In contrast to our work, PINN-DeepONet has only the pre-training phase and suffers from generalization challenges. On the Kolmogorov flow, test-time optimization in our framework  reduces the error from **24.2% to 0.9%** (as shown in Table 2), which is a significant improvement compared to standard operator-learning as in PINN-DeepONet.
> 2. On the architecture level, our framework PINO does function-wise optimization while PINN-DeepONet (trunk net) only optimizes in a pointwise manner (also true for PINN). The function-wise optimization brings significant architecture advantages. For numerical example, on Burgers equation and standard operator learning scenario, PINN-DeepONet achieves **1.38%** relative l2 error while our framework PINO obtains **0.37%**  error (as shown in Table 1), which is a significant improvement.
>
> We are happy to add these explanations to our paper to emphasize the differences with previous works.
>
>
> ### 2. Comparison with more PINNs baseline models
>
> We add a comparison experiment against the Locally adaptive activation functions for PINN [(LAAF-PINN)](https://github.com/antelk/locally-adaptive-activation-functions) and Self-Adaptive PINN [(SA-PINN)](https://github.com/levimcclenny/SA-PINNs), as suggested by the reviewers. For the Kolmogorov flow problem, with Re=500, T=[0, 0.5], averaged over 20 testing instances.
>
>
> Resolution 128x128
> ![128x128](https://i.imgur.com/rpYtbTL.png)
>
>
> **As shown in the figure above, both LAAF-PINN and SA-PINN converge much faster compared to the original PINN method, but there is still a big gap with PINO.** LAAF-PINN adds learnable parameters before the activation function; SA-PINN adds weight parameters for each collocation point. These techniques help to alleviate the PINNs' optimization problem, however, as our results show, the optimization landscape is not altered effectively enough to be competitive with PINO. By using operator ansatz, PINO optimizes in a function-wise manner, making the optimization fundamentally different and, as we have demonstrated, easier.
>
> Note that the contribution of PINO is orthogonal to the above methods. One can apply the adaptive activation functions or self-adaptive loss in the PINO framework as well. **All techniques used to improve PINNs can be straightforwardly transferred to PINO.** We believe it would be an interesting future direction to study how all these methods work with each other for different problems.

---

### Comment · Reviewer_oyPM · 2021-11-18
**Author's response regarding reviewer Tav6's novelty concern?**

I am very curious about the author's response to review Tav6's novlety concern. I would adjust my score down to (3) reject if the authors did not respond in the given discussion time frame as I think novelty is an important consideration for ICLR.

---

> ### Author Response · Authors · 2021-11-20
> **Novelty and contributions**
>
> Thank the reviewer oyPM for the active involvement. Your feedback is valuable to us. Regarding your concern, the [Physics-informed DeepONet](https://arxiv.org/abs/2103.10974) and PINO have different goals and problem settings.
>
> Previous works such as PINN-DeepONet (Wang et. al.) and Physics-constrained DL modeling (Zhu et. al.) use the PDE constraints in operator learning (corresponding to the pre-training phase in PINO.) On the other hand, **the goal of PINO is to use pre-trained operator ansatz to overcome the optimization challenges in PINN and the generalization issue in operator learning,** none of these was addressed in PINN-DeepONet. PINO is the first work, to our knowledge, that utilizes operators ansatz in equation solving, where we propose the **pre-train and optimize** framework, which not just applies PINN-loss to FNO, but fundamentally integrates the two problems settings of operator learning and equation solving. The PINO framework can be extended and applied to other operator models such as DeepONets (if we pre-train the branch net and optimize the trunk net,) which we believe will be an interesting future direction.
>
> Numerically, we do include a comparison between PINN-DeepONet and PINO (just pre-train) on the Burgers equation for learning operators (page 8, first paragraph), where PINN-DeepONet get 1.38% (as reported in the Wang et. al.'s paper) and PINO gets a 0.37% error.
>
> Technically, we propose several methodological advances as well as extensive experiments to understand the optimization and generalization challenges.  Our methodological advances include:
> 1.Instance-wise fine-tuning at test-time to further improve the fidelity of the operator ansatz.
> 2.Novel formulation for inverse problems that result in accurate recovery as well as good speedups.
> 3.Efficient Fourier-space methods for computing derivatives present in the PDE loss.
> 4.Efficient learning through the design of data augmentation and loss functions.
> We now elaborate on each of the above points.
>
> 1. We are the first to formulate PDE learning in two phases: pre-training and fine-tuning. This way we are able to simultaneously overcome the optimization challenges in PINN as well as generalization challenges in operator learning (i.e deeponet or FNO). Also, with this, we can control the accuracy level based on the number of iterations of fine-tuning. Thus, we have a flexible framework where we can tradeoff acuracy with speed for a ML method, like we do with numerical solvers.  In the revised version, we provide these tradeoff plots comparing PINO with other methods including numerical solvers. As shown in Figure 2, PINO has a much better convergence rate compared to PINN.
>
> 2. We are the first to formulate inverse problems as a combination of pre-training and fine-tuning, as well as either forward or backward operator. By directly learning the backward operator, we can obtain the inverse solution directly, without needing to use MCMC with a forward solver. This results in both speedups as well as accurate recovery (in cases like the Darcy flow problem in the paper where the inverse problem is not very ill-posed).
>
> 3. We are the first to develop efficient Fourier-space methods to compute derivatives present in the PDE loss for the Fourier neural operator. In our setting, the input is the full-field (coefficients at all the locations). So computing gradients through autograd is too memory-intensive and infeasible. By developing Fourier-space methods, we make PINO practical for solving challenging PDE families.
>
> 4. We also study the role of augmentation techniques and loss design. Since obtaining ground-truth solutions from numerical solvers is expensive, we augment instances with only PDE loss during pre-training, which can be done without any ground-truth solutions. This improves the generalization capabilities of the operator ansatz. For fine-tuning at test-time on a given instance, we   propose a loss that combines PDE loss of the instance with operator-learning loss from pre-training to avoid overfitting and catastrophic forgetting. While similar methods have been developed for computer vision problems, this is the first time they are studied for PDE problems.

---

### Author Response · Authors · 2021-11-19
**Novelty and contributions**

In this works, we combine operator learning with equation solving (test-time optimization). **Our major novelty and contributions are to use the pre-trained operator ansatz with instance-wise fine-tuning to overcome the optimization challenges in PINN and the generalization challenges in operator learning.**

Previous works such as PINN-DeepONet (Wang et. al.) and Physics-constrained DL modeling (Zhu et. al.) use the PDE constraints in operator learning, as we do during the pre-training phase in PINO. However, we propose several methodological advances as well as extensive experiments to understand the optimization and generalization challenges.  Our methodological advances include:
1.Instance-wise fine-tuning at test-time to further improve the fidelity of the operator ansatz.
2.Novel formulation for inverse problems that result in accurate recovery as well as good speedups.
3.Efficient Fourier-space methods for computing derivatives present in the PDE loss.
4.Efficient learning through the design of data augmentation and loss functions.

We now elaborate on each of the above points.

1. We are the first to formulate PDE learning in two phases: pre-training and fine-tuning. This way we are able to simultaneously overcome the optimization challenges in PINN as well as generalization challenges in operator learning (i.e Deeponet or FNO). Also, with this, we can control the accuracy level based on the number of iterations of fine-tuning. Thus, we have a flexible framework where we can tradeoff accuracy with speed for an ML method, as we do with numerical solvers.  In the revised version, we provide these tradeoff plots comparing PINO with other methods including numerical solvers. As shown in Figure 2, PINO has a much better convergence rate compared to PINN.

2. We are the first to formulate inverse problems as a combination of pre-training and fine-tuning, as well as either forward or backward operator. By directly learning the backward operator, we can obtain the inverse solution directly, without needing to use MCMC with a forward solver. This results in both speedups as well as accurate recovery (in cases like the Darcy flow problem in the paper where the inverse problem is not very ill-posed).

3. We are the first to develop efficient Fourier-space methods to compute derivatives present in the PDE loss for the Fourier neural operator. In our setting, the input is the full-field (coefficients at all the locations). So computing gradients through autograd is too memory-intensive and infeasible. By developing Fourier-space methods, we make PINO practical for solving challenging PDE families.

4. We also study the role of augmentation techniques and loss design. Since obtaining ground-truth solutions from numerical solvers is expensive, we augment instances with only PDE loss during pre-training, which can be done without any ground-truth solutions. This improves the generalization capabilities of the operator ansatz. For fine-tuning at test-time on a given instance, we propose a loss that combines PDE loss of the instance with operator-learning loss from pre-training to avoid overfitting and catastrophic forgetting. While similar methods have been developed for computer vision problems, this is the first time they are studied for PDE problems.

---

> ### Author Response · Authors · 2021-11-20
> **Pre-training: when and why**
>
> #### 1. Pre-training makes previously unsolvable problems solvable
>
> PINNs face several optimization issues: (1) the challenging optimization landscape from soft physics or PDE constraints, (2) the difficulty to propagate information from the initial or boundary conditions to unseen parts of the interior or to future times, and (3) the sensitivity to hyper-parameters selection. For example, in the long-temporal transient flow example (Section 4.1), none of the methods can achieve a reasonable accuracy since it is extremely challenging to propagate the information from the initial condition to future time steps over such a long interval T = [0, 50] just using the soft physics constraint. However, when we pre-train PINO, we can use the learned neural operator ansatz and the operator loss. **This makes optimization during fine-tuning for a given instance much easier.** In this case, the model does not need to propagate information from the initial to a future time, and the method becomes much more robust to the choices of hyperparameters. In the end, pretrained operator ansatz gets a 1.84% error rate, while other methods completely fail to converge.
>
> #### 2. Pretraining is optional for PINO to outperform PINNs
>
> If the problem is simple and the optimization is tractable, then pre-training is not needed. The major advantage for PINO, compared to PINN-based methods, is a much easier to navigate optimization landscape. **The operator ansatz, even when randomly initialized, still has a much better optimization landscape compared to the PINN-based model.** As shown in the experiments for the Kolmogorov flow and the Cavity flow (Section 4), PINO without any pre-training still gets a much better performance compared to PINNs baselines.
>
> The operator ansatz has a better optimization landscape (even without pretraining) because it does function-wise optimization while PINN optimizes in a pointwise manner. The neural operator parameterizes the solution function as an aggregation of basis functions, and hence, the optimization is done directly in function space. Our results show that this makes optimization much easier compared to optimizing a single function as in PINN.
>
> #### 3. Generalization to new distributions
> Standard operator learning as a supervised learning task usually assumes a joint data distribution for the inputs and outputs, which may not be accessible in real-world scenarios. This makes generalization difficult for practical problems and thus constitutes one of the major motivations for our work.  **By introducing fine-tuning at test time, we can handle distributional shifts.** As shown in the Reynolds-transfer experiment Table (Figure [5] and Table [5]), the operator trained from different Reynolds numbers (corresponding to different distributions) can be easily transferred to each other with a small amount of test-time optimization. This transferability relieves the distribution assumption and makes the PINO framework much more flexible. In this sense, the pre-trained operator is not limited to a specific training distribution, but it can be widely applied to similar classes of problems.
>
>
> #### 4. Computation complexity
> In most of our examples, including the Burgers, Darcy, and long-temporal Navier-Stokes equation, pre-training only takes about **1-2 hours** with a single Nvidia V100 GPU. The most expensive training process (Kolmogorov Flow 400+400k) takes no more than **10-20 hours** on a single GPU. Given that the PINN-based methods take about 10 minutes on average for solving each instance, it is worth pre-training if one wants to solve for O(10^2) instances.
>
> We want to emphasize that the pre-training phase can be done offline. Once trained, such pretrained operators can be generalized to a wide class of problems as discussed above. Similar to numerical software, the pre-trained model is distributed publically for everyone to use. Therefore **we would like to consider the training time as part of designing the numerical solver.** Compared to the months and sometimes years that researchers spend crafting numerical solvers, 20 hours of training time is, in fact, an incredible speed-up.

---

> > ### Author Response · Authors · 2021-11-20
> > **New experiments**
> >
> > To support our study, we add three sets of experiments.
> >
> > ### 1. Comparison with more baseline models
> >
> > We add a comparison experiment against the Locally adaptive activation functions for PINN [(LAAF-PINN)](https://github.com/antelk/locally-adaptive-activation-functions) and Self-Adaptive PINN [(SA-PINN)](https://github.com/levimcclenny/SA-PINNs), as suggested by the reviewers. For the Kolmogorov flow problem, with Re=500, T=[0, 0.5], averaged over 20 testing instances.
> >
> > We plot the convergence plots (please see Figure 6 in the revised paper or via the links below). It is also summarized in the table
> >
> > ![Comparison on resolution 64x64x65](https://i.imgur.com/Jgql3xC.png)
> >
> > ![Comparison on resolution 128x128x65](https://i.imgur.com/rpYtbTL.png)
> >
> > | Computation budget (s) | 0.03s | 1s | 10s | 50s |100s |
> > | -------- | -------- | -------- |-------- | -------- |-------- |
> > |PINO| 28.12% | 23.12%| 7.00%| 3.69%    |3.22%
> > |PINN| -| -|66.89% | 47.25% | 39.68%|
> > |LAAF-PINN|- |- | 54.87%| 33.34%| 29.38%|
> > |SA-PINN|- | - | 47.13%| 30.46%| 29.65%|
> >
> > **As shown in the figure and table above, both LAAF-PINN and SA-PINN converge much faster compared to the original PINN method, but there is still a big gap with PINO.** LAAF-PINN adds learnable parameters before the activation function; SA-PINN adds weight parameters for each collocation point. These techniques help to alleviate the PINNs' optimization problem, however, as our results show, the optimization landscape is not altered effectively enough to be competitive with PINO. By using operator ansatz, PINO optimizes in a function-wise manner, making the optimization fundamentally different and, as we have demonstrated, easier.
> >
> > Note that the contribution of PINO is orthogonal to the above methods. One can apply the adaptive activation functions or self-adaptive loss in the PINO framework as well. **All techniques used to improve PINNs can be straightforwardly transferred to PINO.** We believe it would be an interesting future direction to study how all these methods work with each other for different problems.
> >
> >
> > ```
> > Hyperparameter search range:
> > LAAF: n: {10, 100}, learning rate: {0.1, 0.01, 0.001}, depth {4, 6}.
> > SA-PINNs: learning rate {0.001, 0.005, 0.01, 0.05},
> > network width {50, 100, 200}, depth {4, 6, 8}.
> > ```
> >
> >
> >
> >
> >
> >
> > ### 2. Lid-Cavity flow
> > To study PINO on different boundary conditions, we add an example of lid-cavity flow. We assume the no-slip boundary condition where u(x, t) = (0, 0) at left, bottom, and right walls and u(x, t) = (1, 0) on top, similar to Bruneau & Saad [1]. We choose t ∈ [5, 10], l = 1, Re = 500. The main challenge is to address the boundary using the velocity-pressure formulation.
> >
> > We use the velocity-pressure formulation with resolution 65 × 65 × 50 and the Fourier numerical gradient. It takes 2 minutes to achieve a relative error of 14.52%. Figure [4 ] shows the ground truth and prediction of the velocity field at t = 10 where the PINO accurately predicts the ground truth. Note that no pre-training is done on this example.
> >
> >
> > ### 3. Robustness
> > We have added additional experiments to study the robustness of neural operators on the Burgers, Darcy, and Navier-Stokes equation in the full data setting (similar to FNO). We consider two scenarios (1) train with clean data and (2) train with noisy data. The noise is added on the input via
> >
> > 	x = x + e, where e ~ a * max(x) * N(0,1)
> >
> >
> > |         | Training error | Testing (clean) | Testing (10% noise) |
> > |---------|----------------|-----------------|---------------------|
> > | Darcy |        0.6%        |      1.1%           |       1.2%              |
> > | Burgers   |    0.2%            |      0.2%           |       3.8%              |
> > | NS      |      2.4%          |         2.4%        |         3.9%            |
> > | Darcy  (train with noise)|       0.7%         |       1.2%          |      1.2%               |
> > | Burgers  (train with noise) |     2.0%           |      2.0%           |        2.0%             |
> > | NS     (train with noise) |       2.6%         |        2.6%         |          2.5%           |
> >
> > In general, we observe the neural operator is robust, as long as it's not severely overfitting to the training set. The performance of the neural operator degrades when adding >=10% noise. We observe that the Darcy problem is most robust to noise since the inputs have a piecewise constant structure, while the performance on the Burgers and Navier-Stokes problems degrades when training on the clean dataset and testing on noisy dataset. Training with noise will make the model more robust. In this case, there is no domain shift and the operator becomes robust to noise, but it may come with the cost of a small loss in accuracy.

---

> > > ### Author Response · Authors · 2021-11-20
> > > **Updated version of the paper**
> > >
> > > We make the following changes to the updated version:
> > > 1. In appendix B, we add the new experiments of (a) the comparison study of variations of PINNs and (b) the lid-cavity flow.
> > > 2. In appendix C, we add the formulation of Fourier continuation for different geometries.
> > > 3. We update section 4.3 of the inverse problem.
> > > 4. We update figures [2,3,4] and rearrange tables [2,3] from the main text into the appendix.
> > > 5. We rearrange the conclusion and future works into the appendix.

---

### Decision · Program_Chairs · 2022-01-20

**Decision:**

Reject

**Comment:**

The paper proposes the physics-informed neural operator. It combines the operating-learning and function-optimization frameworks, which improves convergence rates and accuracy over traditional methods. While the paper was well written, several reviewers raised their concerns on the novelty of the paper, especially regarding the difference from PINN-DeepONet (Wang et. al.). Following this, there have been a long discussion between the authors and the reviewers, as well as among the reviewers. As a consequence, we think the authors somehow overclaimed their contributions on combining PINN and operator learning, and there are some important references missing and baselines not compared empirically. With this, the conclusion is that we cannot accept this paper in its current form, and we hope that authors can take all the review feedback into consideration and better position the novelty and impact of their work in the future submissions.